# Computational exploration of novel ROCK2 inhibitors for cardiovascular disease management; insights from high-throughput virtual screening, molecular docking, DFT and MD simulation

Iqra Ali[1,2]*, Muhammad Nasir Iqbal[3], Muhammad Ibrahim[2], Ihtisham Ul Haq[4,5,6], Wadi B. Alonazi[7], Abdul Rauf Siddiqi[1]*

1 Department of Biosciences, COMSATS University Islamabad, Islamabad Campus, Islamabad, Pakistan,
2 Department of Biosciences, COMSATS University Islamabad, Sahiwal Campus, Islamabad, Pakistan,
3 Department of Bioinformatics, The Islamia University of Bahawalpur, Bahawalpur, Pakistan, 4 Department of Physical Chemistry and Technology of Polymers, Silesian University of Technology, Gliwice, Poland,
5 Joint Doctoral School, Silesian University of Technology, Gliwice, Poland, 6 Programa de Pós-Graduação em Inovação Tecnológica, Universidade Federal de Minas Gerais, Belo Horizonte, MG, Brazil, 7 Health Administration Department, College of Business Administration, King Saud University, Riyadh, Saudi Arabia

* araufsiddiqi@comsats.edu.pk (ARS); iqraali857@gmail.com (IA)

## Abstract

Cardiovascular disorders are the world's major cause of death nowadays. To treat cardiovascular diseases especially coronary artery diseases and hypertension, researchers found potential ROCK2 (Rho-associated coiled-coil-containing protein kinase 2) target due to its substantial role in NO-cGMP and RhoA/ROCK pathway. Available drugs for ROCK2 are less effective and some of them depict side effects. Therefore, a set of novel compounds were screened that can potentially inhibit the activity of ROCK2 and help to treat cardiovascular diseases by employing *In-silico* techniques. In this study, we undertook ligand based virtual screening of 50 million compound's library, to that purpose shape and features (contain functional groups) based pharmacophore query was modelled and validated by Area Under Curve graph (AUC). 2000 best hits were screened for Lipinski's rule of 5 compliance. Subsequently, these selected compounds were docked into the binding site of ROCK2 to gain insights into the interactions between hit compounds and the target protein. Based on binding affinity and RMSD scores, a final cohort of 15 compounds were chosen which were further refined by pharmacokinetics, ADMET and bioactivity scores. 2 potential hits were screened using density functional theory, revealing remarkable biological and chemical activity. Potential inhibitors (F847-0007 and 9543495) underwent rigorous examination through MD Simulations and MMGBSA analysis, elucidating their stability and strong binding affinities. Results of current study unveil the potential of identified novel hits as promising lead compounds for ROCK2 associated with cardiovascular diseases. These findings will further investigate via *In-vitro* and *In-vivo* studies to develop novel druglike molecules against ROCK2.

**Data Availability Statement:** All relevant data are within the paper, in tables file and in its supporting information file.

**Funding:** We would like to appreciate King Saud University, Saudi Arabia, for funding this work through the research supporting project number (RSP2023R332). The funders had no role in study design, data collection and analysis, decision to publish, or preparation of the manuscript.

**Competing interests:** The authors have declared that no competing interests exist.

## 1. Introduction

Cardiovascular disorders (CVDs) include heart and blood vessels related diseases such as cerebrovascular diseases, coronary heart diseases, hypertension, stroke, arrhythmia and other conditions caused by high blood pressure and cholesterol, diabetes mellitus, physical inactivity, alcohol consumption, smoking, and obesity etc. Aging, stress, microRNAs, and diet also have its own role in CVDs as it causes several changes in human heart [1, 2]. Risk of developing CVDs is three times higher if one's parents had heart diseases and it is most commonly found in men than in pre-menopausal women [3]. Cardiovascular disorders are the leading cause of mortality worldwide and most deaths are due to heart attacks and strokes. In western countries the situation is worse, about sixty-two million people died with the cardiovascular diseases and fifty million people with hypertension. The situation is going to be worse in Pakistan also. According to WHO report 2019, worldwide mortality rate due to CVDs is 32% [4].

Cardiovascular diseases are associated with the dysfunction of NO-cGMP (Nitric oxide-cyclic 3'-5' guanosine monophosphate) pathway (Fig 1) and this pathway act as target for treating different cardiovascular diseases. NO (Nitric oxide) act as relaxing factor due to which recognized as vasodilator, causing smooth muscle relaxation [5]. Calcium calmodulin complex leads to activation of eNOS (endothelial nitric oxide synthase) which oxidizes the L-arginine into L-citrulline to produce NO as well as it can be produced nonenzymatically. NO targets soluble guanylyl cyclase (sGC) by binding to its heam moiety which is in smooth muscle cells and aids in the relaxation of surrounding smooth muscle cells. Activated sGC increases the concentration of cGMP in tissues which induces the relaxation of vascular tone and in this way there is increase in blood flow which lowers the blood pressure [6]. cGMP the secondary messenger, play major role in cardiovascular system even minor disturbance in any step leads to malfunctioning of cardiovascular system such as hypertension, cardiac hypertrophy, myocardial infarction as well as the heart failure [7]. cGMP also known as cGK (cGMP dependent protein kinase) that found in wide range of eukaryotes [8]. cGKI is most often expressed cGK in cardiovascular systems while cGKII not expressed in cardiac and vascular myocytes [9]. There is calcium dependent and independent contraction and relaxation in cGMP/cGKI pathway when activated MLCK phosphorylate MLC and MLCP dephosphorylate MLC respectively [6].

For the treatment of cardiovascular diseases such as hypertension, cerebral ischemia, vascular inflammation, arteriosclerosis, atherosclerosis, coronary vasospasm, and stroke, ROCK2 was used as therapeutic drug target because ROCK2 inhibit myosin binding subunit (MBS) of MLCP to regulate MLC's phosphorylation at Ser19 residue instead of dephosphorylation and cause calcium independent contraction instead of relaxation. Therefore, both MLCK and MLCP phenomena promote contraction and lead towards different CVDs (Fig 2). If ROCK is inhibited, MLCP promote dephosphorylation of MLC and cause relaxation and in this way rate of hypertension will be reduced [10]. ROCK2 relate to coronary artery diseases and hypertension as well as it plays vital function in the contraction of smooth muscle cells than ROCK1 [11]. Genes of both ROCK1 and ROCK2 have different subcellular localizations where they expressed. Except brain and muscles, ROCK1 is widely expressed in the liver, kidney, lung, spleen, and testis. While ROCK2 is abundantly expressed mostly in brain, heart, muscles, and placenta. All these results confirm the fundamental role of ROCK2 in cardiovascular diseases [12]. So, a significant target ROCK2 was employed for the treatment of numerous cardiovascular diseases i.e., hypercontraction, atherosclerosis, hypertension, and heart failure [13].

ROCK2 (Rho-associated coiled-coil-containing protein kinase) is a serine/threonine protein kinase that is downstream effector of RhoA. ROCK2 also known as p164 ROCK-2 or ROKa, ROCK-II and consist upon 1388 amino acids [14]. It is positioned on chromosome 2 and consists upon three major domains; at the N-terminus of ROCK2 there is a catalytic kinase

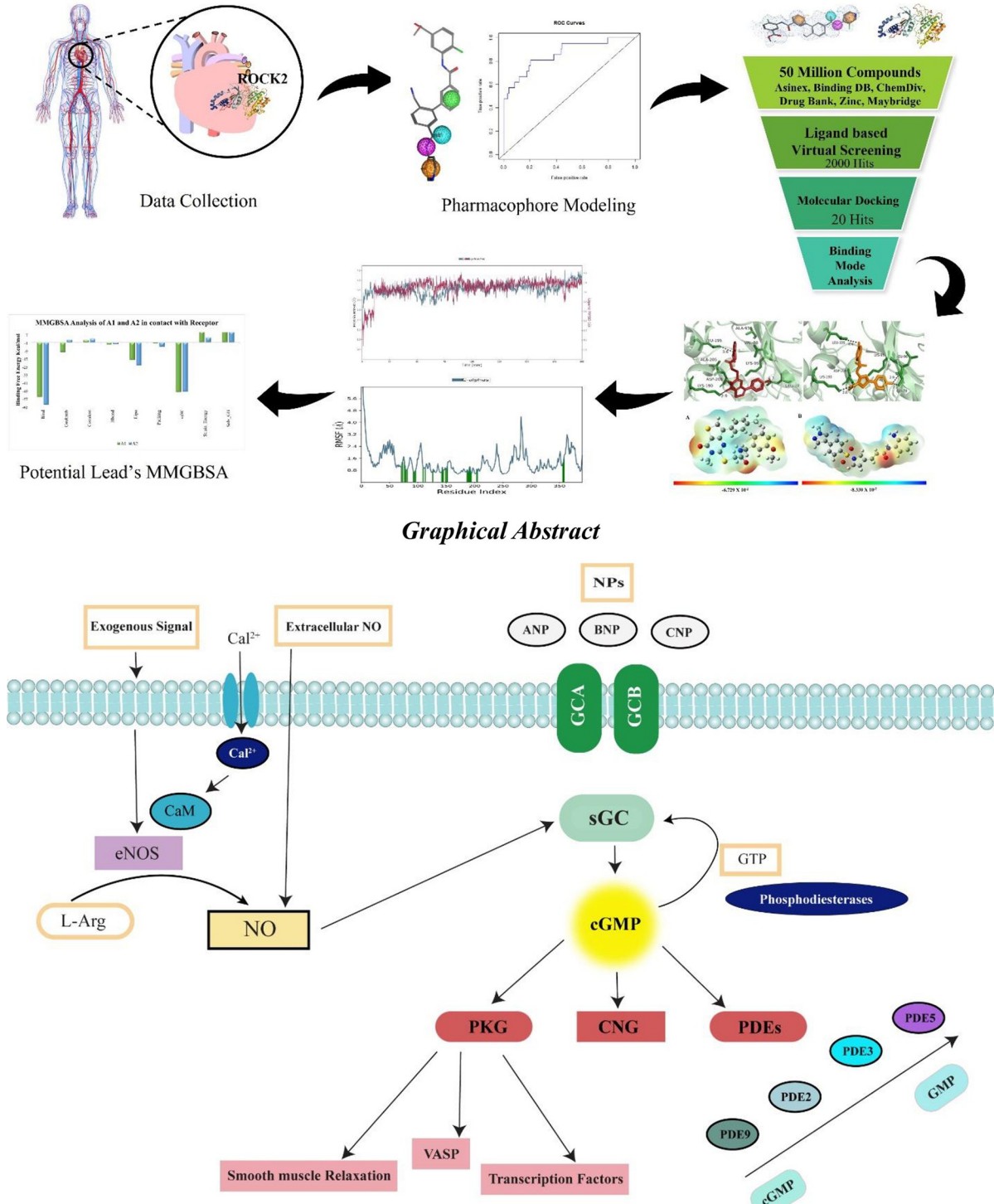

**Fig 1. Human NO-cGMP (Nitric oxide-cyclic 3'-5' guanosine monophosphate) pathway where Ca$^{2+}$ binds with calmodulin and makes a complex that activates eNOS to produce NO by oxidizing L-arginine to L-citrulline.** NO targets sGC to increase the production of cGMP which starts a cascade.

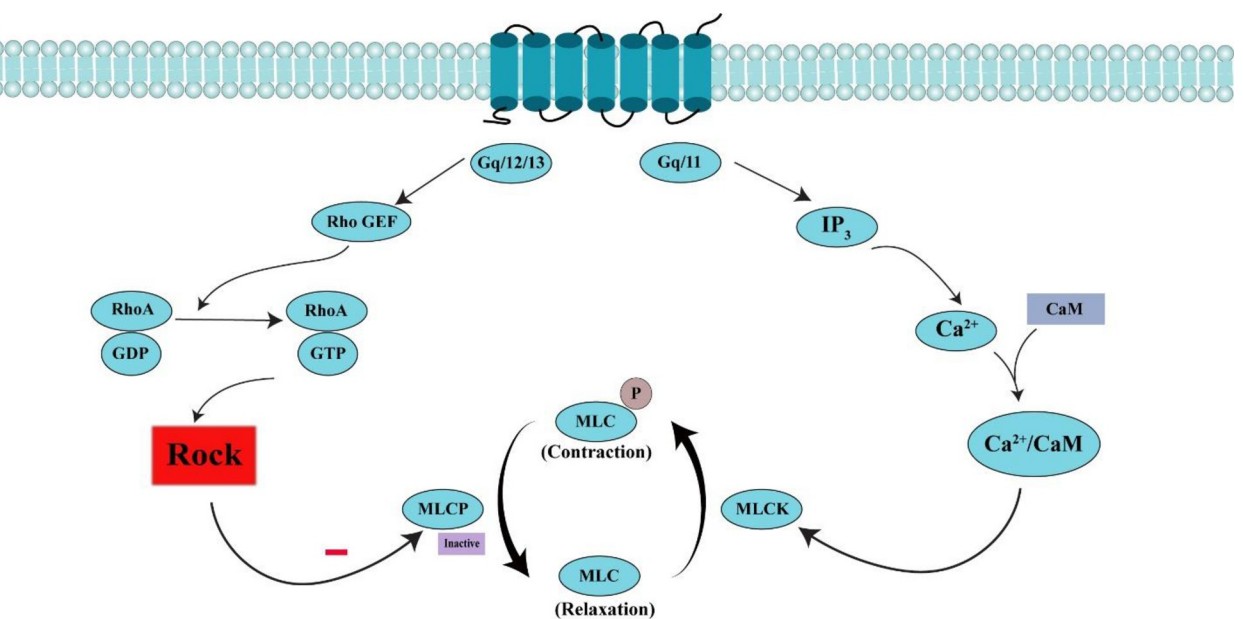

**Fig 2. RhoA/ROCK pathway in cardiovascular system.** Contractile agonist (Gq/11) produces 1P3 which binds to its receptor and releases calcium. Calcium/calmodulin complex activates MLCK that phosphorylate MLC and performs calcium dependent contraction. Gq/12/13 agonist activates RhoA-GDP to RhoA-GTP. Activated RhoA activates ROCK that inhibits MLCP. So, inactive MLCP is unable to dephosphorylate MLC and causes calcium independent contraction instead of relaxation.

domain. After that there is a coiled-coil domain with Rho-binding domain and at the end there is cysteine-rich putative pleckstrin homology domain (PH) present at the C-terminal [15]. Protein's C-terminus inhibitory PH region automatically inhibits N terminal kinase domain and turns into close conformation (inactive). Activated RhoA, activates ROCK by interacting with ROCK's coiled-coil domain and brings conformational alteration (open conformation). (Fig 3) Inactivated ROCK can be activated via Rho-independent activation when arachidonic acid bind to the PH domain or cleavage of the C terminus by caspase-2 or -3, and by granzyme B [16]. This conformational alteration leads to kinase activation [13].

Fasudil, Y27632 and ripasudil are significant ROCK inhibitors that have been endorsed for the treatment of hypertension, cerebral vasospasm, and glaucoma respectively. In addition, there are Indazole and amidoindazole available as ROCK inhibitors to treat breast cancer. DW1865 is another ROCK inhibitor which has shown good cell potency in fiber formation. Dual ROCK1 and ROCK2 inhibitors have been linked to problematic side effects as well as some investigators also hypothesized that ROCK inhibitors would be less effective for the treatment of cardiovascular diseases [17]. Because of these issues it is necessary to predict novel compounds with stronger binding affinity for ROCK2 to treat CVDs. High selectivity for ROCK2 could result into a product candidate with improved tolerability which could allow for long-term systemic use [18].

## 2. Material and methods

The graphical workflow of adopted methodology was illustrated in Fig 4.

### 2.1 Structure refinement and evaluation

X-ray crystallographic structure of targeted receptor was fetched from RCSB PDB depicting 0.281 R-value free and 2.79 Å resolution while amino acid sequence of target protein was retrieved from UniProt database in FASTA format and BlastP was run to determine phylogeny

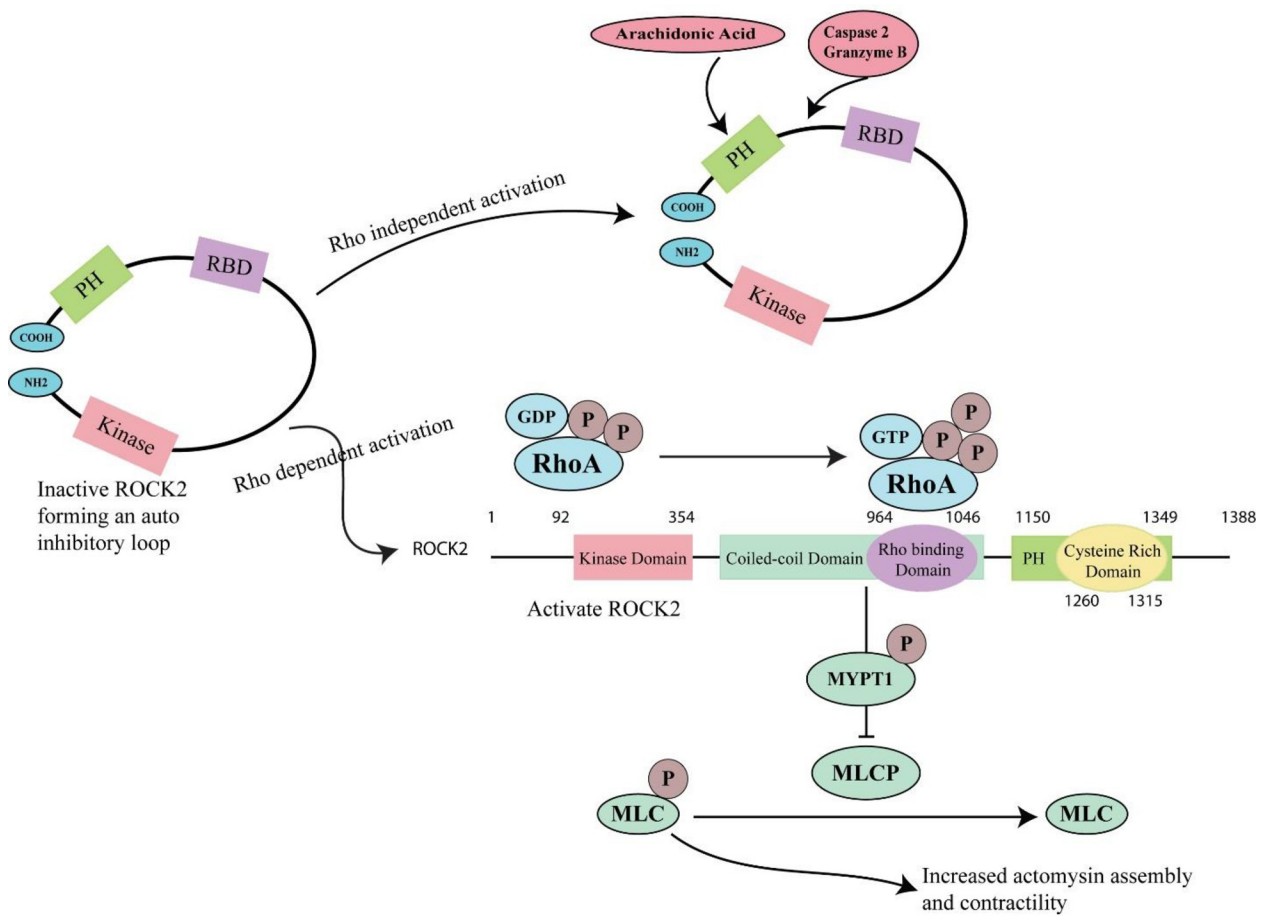

**Fig 3. ROCK2 activation mechanism via RhoA independent and dependent ways.** In RhoA-independent way, arachidonic acid binds to PH domain or Caspase 2/Granzyme B cleaves in between the RBD and PH domain while in RhoA-dependent way, activated RhoA (RhoA-GTP) binds with the Rho binding domain to activate ROCK2 that further phosphorylate MYPT1 subunit of MLCP and resultantly MLCP become inactive. So, MLCP is unable to dephosphorylate MLC and increase the actomyosin assembly and contractility.

of query protein (ROCK2). The Expected value, identity, similarity, and query coverage values help to identify evolutionary relationship between different sequences as well as members of gene family. Expasy ProtParam tool [19] was used to analyze target protein's physiochemical properties such as molecular weight, theoretical pI, atomic composition, instability index, GRAVY (grand average of hydropathicity), and amino acid composition. Domains are accountable for a specific function, which contributes to the overall role of a protein. InterPro database analyses protein sequences functionally by categorizing them into families and anticipates the existance of domains, repeats, binding, and active sites [20]. Prediction of protein subcellular localization is an essential step to understand protein function. To anticipate subcellular localization of eukaryotic proteins, LocTree3, DeepLoc-1.0, ESLpred, and HSLPred tools were used which worked on deep neural networks relying solely on their amino acid composition, dipeptide composition, and physicochemical properties. CASTp [21] was used for predicting active site residues that were further validated by site finder function.

Missing residues were added by utilizing UCSF Chimera's built-in discrete optimized protein energy (DOPE) loop modeling protocol and evaluated modeled structure on the bases of zDOPE and GA341 score. GalaxyRefine, and 3Drefine were employed for refinement of loop and terminus regions using ab initio modeling combined with atomic-level energy

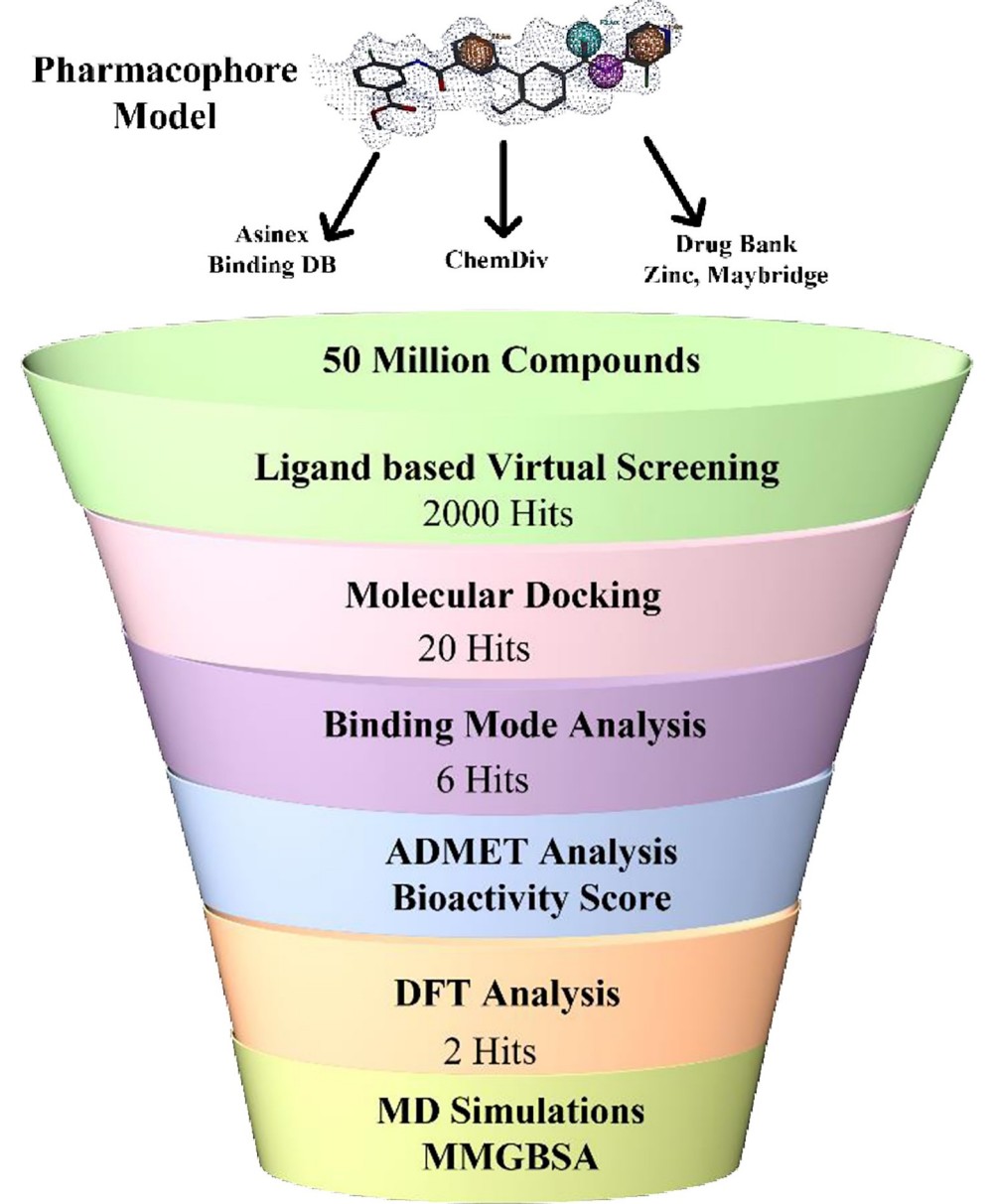

**Fig 4. Graphical workflow of research.**

minimization. SAVES Server (ERRAT, Verify3D, PROVE, and PROCHECK) was used to validate and examine the stereochemical quality of protein structure. ERRAT and VERIFY 3D distinguishes between correctly and incorrectly folded regions of protein structure [22, 23]. PROCHECK performs a thorough examination of protein structure's stereochemistry as well as draws attention to regions of protein that may require further investigation while PROVE checks the deviation using standard parameters [24].

## 2.2 Pharmacophore query generation and validation

55 co-crystal structures of ROCK2 were fetched from PDB (S3 Table in S1 File), and 18 active compounds were selected based on IC$_{50}$ values and their physiochemical properties. Common

features were found to build pharmacophore query that depict bioactivity towards target protein. Between two pharmacophoric features, a maximum 10 Å distance was allowed. Validation of the pharmacophore model was accomplished via Receiver Operating Characteristic curve (ROC) yielded area under curve (AUC) and enrichment factor. For this purpose, 18 co-crystal compounds were used as an active control, and a decoy dataset of 619 compounds from ChEMBL database was utilized as a negative control. Physiochemical properties were set as a threshold to filter chemically distinct compounds from ChEMBL and generated 10 conformers of each decoy molecule using Avogadro [25]. Target protein docked with active and decoy molecules and docking scores were further used to generate ROC curve. AUC access test's overall accuracy and values ranges from 0–1, where 0 shows perfectly inaccurate test and 1 denotes thoroughly accurate test. An AUC of 0.5 (black diagonal line) indicates no differentiation, 0.7–0.8 deemed acceptable, 0.8–0.9 considered excellent, and values beyond 0.9 as remarkable.

### 2.3 Pharmacophore-based virtual screening of prepared library

50 million drug-like and lead-like chemical compounds were retrieved from different databases like Asinex, Binding Database, Chembridge, ChemDiv, Drug Bank (FDA-approved molecules), Maybridge, and Zinc Database. Pharmacophore query employed to screen prepared libraries via virtual screening protocol. Virtual screening was carried out to locate hit compounds that share chemical and structural similarities with pharmacophore model and could inhibit target protein's activity.

### 2.4 Molecular docking

Screened compounds were subjected to molecular docking in the active site of ROCK2 to find out best binding pose under physiological conditions (300K, PH 7, 1 atm, 0.1 salt concentration), and for this purpose detailed conformational analysis was undertaken employing multiple docking algorithm and molecular graphics program were used for good quality 2D visualizations. It provides results on the basis of empirical and force field-based scoring functions. Both receptor and hit compounds were prepared by protonation using Protonate3D module and energy minimization by using Amber ff10 and EHT (Extended Hückel Theory) force field for macromolecules and ligands respectively. AMI-BCC charges are projected for small molecules as well adjust hydrogens and lone pairs as required. Both protein and hit compounds were saved into their respective file formats then site-specific docking was processed by implementing triangle matcher protocol with rigid receptor docking method. London dG the initial scoring method evaluates initial 30 ligand placements while GBVI/WSA dG (generalized-born volume integral/weighted surface area) scoring function refines best 5 poses. The docked poses of ligands were further evaluated via binding mode, S score, and RMSD values. PyMOL was used for 3D visualization of receptor-ligand while Discovery Studio for 2D depiction.

### 2.5 Evaluation of pharmacokinetic properties

A successful lead needs to be more druglike for successful drug discovery so top compounds were screened based on drug-likeness, ADME (Absorption, Distribution, Metabolism, and Excretion), and toxicity characteristics through PkCSM, ADMETlab, and SwissADME which calculate important drug like descriptors as well as used for predicting mutagenicity, immunotoxicity, hepatotoxicity, and carcinogenicity. The druglike compounds should have to follow certain rules such as Lipinski's, Veber's, Ghose's, Egan's, and Muegge's rules. These rules trace druglike molecules on the basis of 2D depiction (physical, chemical properties) and molecules

must have to fulfill up to 2 of these rules otherwise represent some ADMET obstacles. Drug likeness is an important concept in drug development as it helps in identifying compounds of good pharmacokinetic properties [26]. These approaches help to reduce the number of compounds that need to be tested in clinical trials as well as save time and resources in drug development. After ADMET and drug-likeness analysis, we will have compounds whose chances of selection in clinical trials increases.

## 2.6 Bioavailability and bioactivity score prediction

Bioactivity and bioavailability scores are significant factors in drug discovery/development and depict overall potential of lead compounds. These factors help researchers to prioritize and optimize compounds for further stages of evaluation. Bioavailability determines compound's effectiveness via physicochemical properties, such as solubility and permeability via SwissADME. To predict biological activity of selected compounds, Molinspiration (www.molinspiration.com) software was employed. Drug molecule's bioactivity can be impacted by several parameters, including its chemical structure, potency, and selectivity for a particular target. The drug is expected to bind primarily with target proteins i.e., enzymes, ion channels, and receptors. A drug molecule with high potency and selectivity for a particular target is more likely to deliver a therapeutic benefit with fewer side effects than a molecule with lesser potency or selectivity [27]. Bioactivity score of top 6 complexes were predicted against human receptors such as GPCR (G protein-coupled receptors), kinases, ion channels, proteases, and enzyme inhibitors.

## 2.7 DFT analysis

To analyze the molecular geometry, conformation, energetics and reactivity of top selected compounds, density functional theory (DFT) analysis was performed via Gaussian 09 and GaussView 06 package. HOMO and LUMO energies were estimated to calculate the energy gap between the highest occupied and lowest unoccupied molecular orbitals. To identify electrophilic and nucleophilic functional sites of A1 and A2 compounds, MEP (molecular electrostatic potential) was carried out. For this, top compound's geometries were optimized by applying B3LYP method with 6-3IG** basis set. To figure out stability and reactivity of A1 and A2 compounds, HOMO, LUMO, energy gap ($E_g$ = ELUMO—$E_{HOMO}$), electronegativity, electrophilicity index, electrochemical potential, hardness and softness parameters were computed.

## 2.8 MD simulation

200 ns MD simulation was employed to study protein-ligand interactions at atomic level and conformational changes during binding process by applying Newton's laws of motion via Desmond from Schrodinger suite [28]. Both systems (A1 and A2 complexes) were preprocessed via protein preparation wizard module in Maestro using default settings while whole system was set up by "system builder tool". Minimize energy of both systems then equilibrate in the orthorhombic box (10Å×10Å×10Å) of TIP3P water model, neutralized by adding 0.15M NaCl concentration and heating gradually from 200K to 250K and then 300K with 1 atm pressure [29]. Receptor-ligand interactions were examined via simulation interaction diagram tool. The MD simulation was successfully conducted using Dell T7810 Precision Workstation with an Intel Xeon E5-2670 v3 processor, 64 GB RAM, and Zotac GeForce RTX 3070 GPU on a system running Ubuntu 20.04.1 LTS.

## 2.9 MMGBSA calculations

MM-GBSA (molecular mechanics generalized Born surface area) analysis was conducted via prime module of Schrödinger to evaluate binding free energy ($\Delta G_{bind}$) of both systems (ROCK2 complexed with A1 and A2) [30] throughout the simulation time. Counter ions were stripped, VSGB solvent model with OPLS 2005 force field were employed along rotamer search techniques to calculate $\Delta G_{bind}$. The calculations were based on MD trajectory frames taken at 10ns intervals after MD run. **Eq (1)** The total binding free energy is the difference between the energy of protein-ligand complex ($G_{complex}$) and free energy of individual protein ($G_{protein}$) and inhibitor ($G_{ligand}$). $\Delta G_{bind}$ calculated via **Eq (2)**, a combination of molecular mechanics gas phase energy ($\Delta E_{gas}$), solvation free energy ($\Delta Gs_{ol}$), and entropy (T$\Delta$S) [31].

$$\Delta G_{bind} = G_{complex} - (G_{protein} + G_{ligand}) \tag{1}$$

$$\Delta G_{bind} = \Delta E_{gas} + \Delta Gs_{ol} - T\Delta S \tag{2}$$

$$\Delta E_{gas} = \Delta E_{int} + \Delta E_{ELE} + \Delta E_{VDW} \tag{3}$$

$$\Delta Gs_{ol} = \Delta G_{polar} + \Delta G_{nonpolar} = \Delta G_{GB} + \Delta G_{S} \tag{4}$$

$$T\Delta S = T(\Delta S_{trans} + \Delta S_{rot} + \Delta S_{vib} \tag{5}$$

Gas phase interaction energy ($\Delta E_{gas}$) is equal to van der Waals, and electrostatic energy while internal energy is neglected (Eq (3). (Eq (4) Solvation free energy ($\Delta Gs_{ol}$) is the sum of polar (generalized born) and nonpolar energy (solvent accessible surface area) [32]. T$\Delta$S denotes conformational entropy equivalent to translational, rotational and vibrational entropy changes upon binding (Eq (5)).

## 3. Results and discussion

The accession number, PDB, and UniProt ID of target protein are NP_004841.2, 6ED6, and O75116 respectively. ROCK2 has Cyclic—C2 global symmetry while global stoichiometry is Homo 2-mer—A2, and physiochemical properties are listed in S1 Table in S1 File. Missing residues were added, and model with lowest zDOPE score (-1.16) and highest GA341 score (1.000) was chosen as the best one. Fig 5 depicts 3D structure of receptor along the Ramachandran plot and highlights amino acids that are in disallowed regions (white color). According to Ramachandran plot, 95.4% of residues lied in the core regions (red color), 4.3% of residues in additional allowed region (yellow) of torsion values, and 0.3% of residues in generously allowed regions (light yellow) with no amino acid belongs to active site of ROCK2. (S1A Fig in S1 File). Overall quality factor of protein was 93.0748%, 0.26 Å RMSD, and 1.5 MolProbity score indicating physical correctness of all atoms within protein structure. S1B Fig in S1 File. represents correctly and incorrectly folded regions of protein and score ranges from -1 to +1. Approximately, 89.26% amino acids have averaged score > = 0.2 and 0.471 Z-score.

S2 Fig in S1 File depicts domains and functional sites of receptor and the main catalytic kinase domain of ROCK2 illustrated in pink color from residue 92 to 354. DeepLoc-1.0, ESLpred, and HSLPred web servers predict that target protein is present in cytoplasm of cell with 0.653 likelihood and 97% accuracy (S3 Fig in S1 File.). Binding pocket residues of receptor identified through CASTp and site finder function are labeled and displayed in Fig 5A. S2 Table in S1 File represents binding pocket residues of ROCK2.

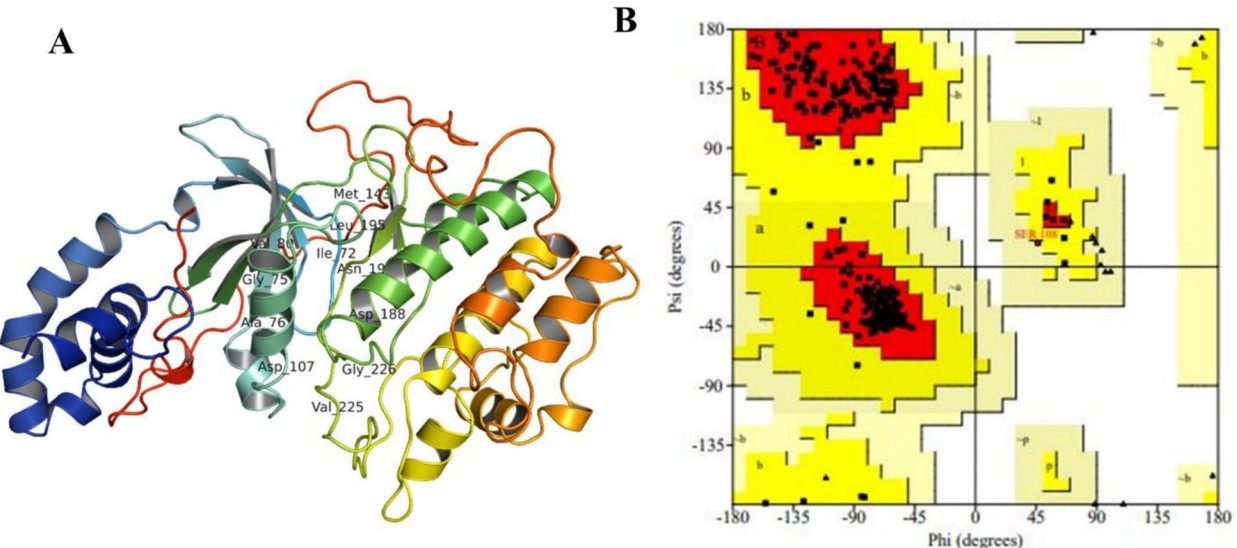

**Fig 5.** (A) 3D structure of ROCK2 with labeled active site residues along the (B) Ramachandran Plot exhibiting different regions specified with different colors. Most favored regions with red color, additional allowed regions with yellow color, generously allowed regions with light yellow color, and disallowed regions with white color.

## 3.1 Pharmacophore quality assessment via ROC curve

Ligand-based pharmacophore model contains four pharmacophoric features i.e., F1 and F4 for aromatic functional group, one nitrogen as HBD (F2), one oxygen as HBA (F3) on the basis of PCH-All scheme with 1.2 tolerance, 50% threshold and 100% score (weighted conformation) (Fig 6A). The selected features have defined distances as 2.79 Å for AroF1-HDrF2, 2.22 Å for HDrF2-HAcF3, 6.21 Å for HAcF3-AroF4 and 9.77 Å for AroF4-AroF1. These features help to find out similar/query-like molecules and comparable binding modes. Aligned view of compounds used to build the query depicted in S4 Fig in S1 File. Docking scores of 18 active and 6192 decoy compounds were carried out to build ROC curve for validation of

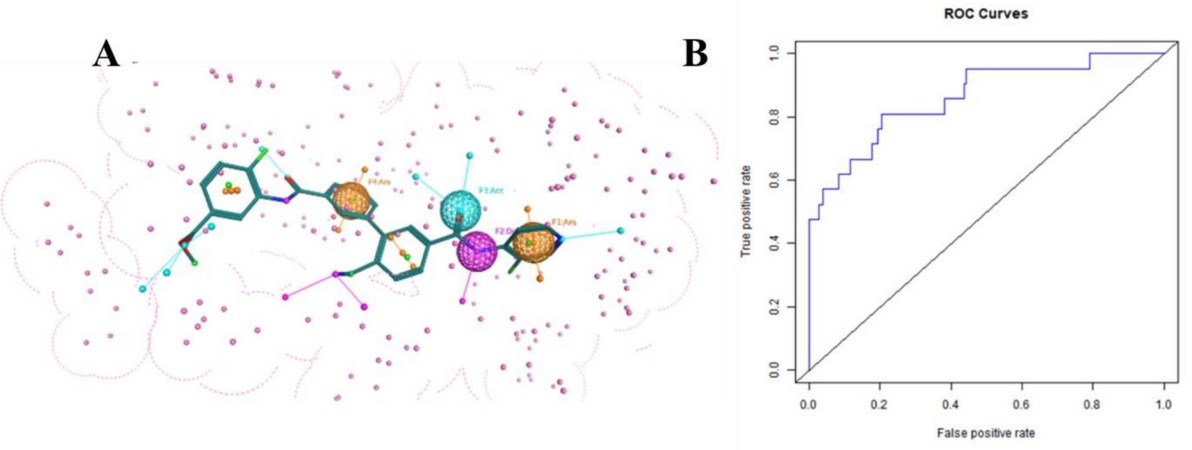

**Fig 6.** (A) Pharmacophore model with common highlighted features in sphere form. F represents number of features where Aro stands for Aromaticity, Acc represent hydrogen bond acceptor, Don shows hydrogen bond donor. (B) ROC_AUC graph for Pharmacophore query validation.

pharmacophore model. Fig 6B depicts an acceptable area under the curve (AUC) 0.862 means 86% chance that modeled query will select active molecules instead of inactive with satisfactory enrichment factor of top 1% (43.095) and 20% (3.727).

A validated pharmacophore model was used to perform virtual screening of prepared library of ∼50M compounds that yielded 5540 compounds. Then these compounds were filtered on the basis of LIPINISKI'S "rule of five" (MW<500, HBD< 5, HBA< 10, logP < 5), and 2085 hits were retained. Further compounds were screened based on hit rate (80%) and a cutoff of RMSD values of more than 1.5Å. Finally, 2000 best hits retrieved will likely bind with the same active site of ROCK2.

## 3.2 Binding poses and interactions

Molecular docking was performed to further screen hit compounds and establish binding modes of selected molecules that may help to treat the disease by inhibiting ROCK2. Before this, docking protocol validated via ROC curve which depicts false positive and true positive fractions on x-axis and y-axis respectively (S7 Fig in S1 File). Based on receiver operating characteristic curve (ROC curve), 0.853 AUC (area under curve) value was observed, and enrichment factor in top 1% was observed 8.88. The results suggest that docking protocol did not produce false-positive results.

Docking of standard drugs i.e., Fasudil, Y27632, ripasudil and cocrystal compound 3SG illustrate that they make major interactions with Asp206, Val80, and Lys95 residues and bind in the active site of ROCK2 (S5 Fig in S1 File.). Docking cut-off for potential hits was based on score-based ranking of binding affinity and RMSD as ≥ -11 kcal/mol and ≤ 1.1 Å respectively. Fig 7 represents top 6 hit compounds while Fig 8 revealed interaction mechanism of top-ranked poses of 6 hits which bind in the same binding site as the reference compound 3SG. A1 demonstrated best binding score (-14.2884 kcal/mol) with 1.09Å RMSD among other 14 compounds. Positively charged Lys190 makes polar hydrogen bond with A1 (2.9 Å distance) while Ala93, Val80, Leu97, Ala205 make pi-alkyl interactions. Nonpolar, aliphatic Leu195 made pi-sigma interactions and positively charged Lys95 make pi-cation interactions (Fig 8A). Upon comparative analysis with the co-crystallized ligands, the pivotal interactions established by co-crystallized ligands with ROCK2 predominantly involve amino acid residues Lys190 and Asp206. Leu195, negatively charged Asp206, Lys95 illustrate alkyl, pi-lone pair and salt bridges respectively with A2 compound. Lys190 makes polar hydrogen bond with 2.6 Å distance. Gly78, Leu97 make pi-sigma and pi-alkyl interactions with 2.2 and 2.9 Å (Fig 8B). A3 made 3 hydrogen bonds with Asp107, Lys190 and Leu96 lie within 2.5, 2.8 and 3.3 Å distance. Also, it made pi-anion and cationic interactions with Asp206 and Lys95 while Gly75 depicts pi-sigma (Fig 8C). A4 displays carbon hydrogen bond with Glu144 and pi-donor hydrogen bond with negatively charged Asp192 and Asp206 (2.9, 2.9 and 3.2Å). Met146, 118, Val152, 127, 80, Ala93, Phe358 depict pi alkyl interactions while Leu195 display pi-sigma interaction (Fig 8D). A5 shows binding affinity of -11.38 kcal/mol with polar hydrogen bonds of Asp192, Asp188, and Gly75 (Fig 8E). A6 depicts polar hydrogen bond with negatively charged Glu114, pi-sulfur and pi-alkyl interactions with Met143, aromatic Tyr145 (3.1Å) and Phe358 (3.7Å) (Fig 8F).

Ligand efficiency is a useful parameter to select lead compound and in the optimization process. Table 1 depicts ligand efficiency, RMSD, and different types of interactions ligand made with receptor along their distances. Fig 9 shows comparison of binding affinity and RMSD of first 6 hit compounds along their correlation analysis which depict interdependence of ligand efficiency and drug score. Compounds were further analyzed by superimposing them on reference compound and S6 Fig in S1 File. illustrates superimposed view of reference compound with two top hits and their surface mapping along 2D depiction. On the basis of

**Fig 7. Top hit compounds obtained based on binding mode, binding affinity, and RMSD values.**

results, it could be anticipated that both A1 and A2 compounds favorably bind to the active site of ROCK2 and particularly Lys190, Leu195, and Asp206 residues play significant role to inhibit the activity of ROCK2 receptor.

### 3.3 ADME analysis, drug likeness, and toxicity assessment

Poor ADMET profile is responsible for the dropout of lead compounds during clinical trials. So, it's necessary to trace these issues at the early stages of drug discovery. For this, top 6 compounds carried out for ADME and toxicity analysis. All compounds lie within permissible range specified in RO5 (HA, MW, HBD, HBA, and Log P) and no compound violates Veber, Egan, and Muegge's guidelines. As the successful lead must satisfy the ADMET properties. Here A1 and A2 compounds satisfy ADMET properties along synthetic accessibility score of 3.59, 3.47 with high GI absorption and no blood-brain barrier (Table 2). Fig 10 depicts heatmap of ADME properties of top six compounds ranging from low (blue) to high (yellow).

According to toxicity profile analysis, compounds didn't show tumorigenic, mutagenic, and reproductive effects. As well they didn't show skin sensitization and irritation. Toxicity analysis of selected compounds were enlisted in Fig 11A while Fig 11B depicts distribution and correlation between oral rat acute toxicity and carcinogenicity. According to Fig 11, top A1 and A2 compounds have acceptable toxicity profile, and can be used as lead compounds for further analysis.

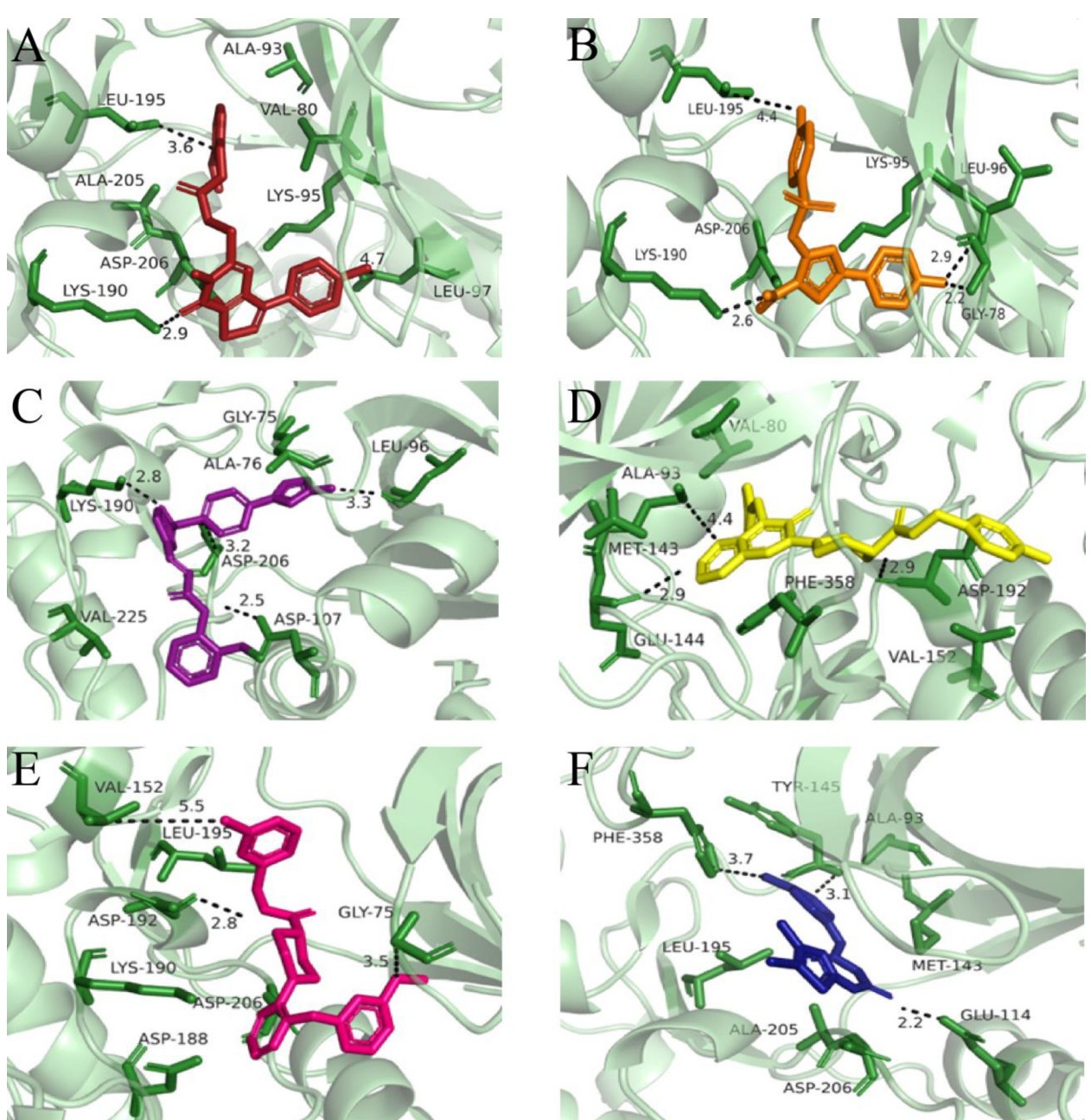

**Fig 8. The interactions of first 6 docked compounds with the key residues of ROCK2 binding pocket.** (A) A1 (B) A2 (C) A3 (D) A4 (E) A5 (F) A6 compound. The interactions denoted with black dash lines and distance between the compounds and binding site residues measured in Å.

### 3.4 Bioactivity score prediction

The lead compound with high bioavailability score will be able to reach its target site in adequate amounts to exert therapeutic effect. Bioactivity scores provide details of drug binding for various receptors such as nuclear receptor ligand, protease, and enzyme inhibitors. Table 3 illustrates bioactivity scores, and all compounds lie within -5.0 to 0.0, showing moderately active compounds while some depict scores above zero illustrating the good activity of compounds. The values clearly indicate that lead compounds possess properties required to act as potential drugs.

**Table 1. Binding affinity, RMSD scores, interaction analysis and some other calculated features of hit compounds.**

| I Ds | Binding Affinities (kcal/mol) | RMSD (Å) | Interacting Residues | Type of Interaction | Distance (Å) and Angle (H-bonds) | Ligand Efficiency (kcal/ mol/HA) |
|---|---|---|---|---|---|---|
| ChemDiv_F847-0007 | -14.2884 | 1.0908 | Lys190, Asp206, Leu195, Leu97, Lys95, Val80, Ala93, Asp206 | Polar hydrogen bond, Pi-sigma, Pi-alkyl, Pi-cation, Alkyl, Pi-alkyl, Pi-anion | 2.9 (103.4°), 2.53 (102.4°), 3.35, 4.6, 4.19, 4.6, 5.02 & 4.8, 3.08 | -0.446 |
| DrugBank_9543495 | -12.3278 | 1.0570 | Lys190, Asp107, Leu195, Asp206, Lys95, Gly78, Arg74, Leu96, Leu97 | Polar hydrogen bond, Alkyl, Pi-lone pair, Salt bridge, Pi-sigma, Amide pi-stacked, Halogen (fluorine), Pi-alkyl | 2.6 (111.4°), 2.48 (115.4°) 4.4, 2.99, 5.20, 2.2, 5.13, 2.9, 5.5 | -0.474 |
| DrugBank_15985904 | -12.0973 | 1.0956 | Asp107, Lys190, Asp206, Leu96, Val225, Lys95, Gly75 | Polar hydrogen bond, Polar hydrogen bond, Pi-anion, Carbon hydrogen bond, Pi-alkyl, Pi-cation, Pi-sigma | 2.5 (103.4°), 2.8 (103.4°), 3.2, 3.3 (116.4°), 5.4, 4.9, 3.96 | -0.378 |
| ChemDiv_50834432 | -11.5857 | 1.0700 | Glu144, Met143, Met146, 118, Val152, 127, 80, Leu195, Ala93, Phe358, Asp192, 206, Leu195 | Carbon-hydrogen bond Pi-Sulfur, Pi-Alkyl, Pi-Alkyl, Pi-sigma, Pi-Alkyl, Pi-Alkyl, Pi-donor, hydrogen bond, Pi-Sigma | 2.9 (105.4°), 3.40, 5.43 & 5.35, 4.65, 5.06, 4.50, 4.5, 4.4, 5.3, 2.9 & 3.23, 3.6 | -0.373 |
| ChemDiv_50834372 | -11.3850 | 1.0921 | Asp192, Val152, Lys190, Asp206, Asp188, Gly75, Leu195 | Polar hydrogen bond, Alkyl, Pi-anion, Pi-cation, Carbon hydrogen bond, Carbon hydrogen bond, Pi-alkyl | 2.8 (109.4°), 5.5, 4.03 4.5, 2.32 (116.4°), & 2.47 (113.4°), 3.5, 5.4 | -0.355 |
| ChemBridge_2805917 | -11.0052 | 1.6322 | Glu114, Asp206, Ala205, Met143, Ala93, Leu195, Tyr145, Phe358 | Polar hydrogen bond, Attractive charge, Alkyl, Pi-sulfur, Alkyl, Pi-alkyl | 2.2 (103.3°), 4.3, 5.2, 5.7, 3.6, 4.7, 3.1, 3.7 | -0.524 |
| ZINC000340512830 | -12.4738 | 1.9142 | Asp192, Val127, Met118, Ala205, Leu195, Ala93, Val80 | Carbon hydrogen bond, Alkyl, Pi-sigma, Pi-alkyl | 2.01 (105.4°), 5.2, 4.6, 3.4, 3.42, 4.6, 4.63 | -0.498 |
| ZINC0003343621432 | -11.9135 | 0.9396 | Asn193, Ala93, Met143, Asp150, Met118, Val80, Leu195, Ala205, Val127, Met146 | Carbon hydrogen bond, Pi-alkyl, Pi-anion, Pi-alkyl, Pi-sigma, Pi-alkyl | 3.4 (102.6°), 4.1, 4.5, 3.2, 4.7, 5.2, 3.5, 3.7, 5.3, 5.2 | -0.794 |
| ZINC000156970545 | -11.4903 | 0.7396 | Met146, Ile72, Ala93, Met118, Val127, Met143, Asp206, Leu195, Ala205 | Polar hydrogen bond, Pi-alkyl, Pi-sulfur, Pi-alkyl, Pi-anion, Pi-sigma | 2.4(107.5°), 5.3, 3.7, 5.5, 5.3, 5.06, 4.04, 3.8, 2.8 | -0.718 |
| ZINC001552885737 | -11.4977 | 1.8301 | Asp206, Tyr145, Met146, Asp192, Asn193, Val80, Ala93, Arg74, Ile72, Phe358, Ilu195 | Polar hydrogen bond, Alkyl, Carbon hydrogen bond, Pi-alkyl, Carbon hydrogen bond, Pi-alkyl, Pi-sigma | 3.09 (132.4°), 4.3, 4.5, 2.64 (113.9°), 2.8, 4.9, 3.8, 2.34 (110.5°), 4.8, 4.9, 3.12 | -0.459 |
| MayBridge_2800050 | -11.1272 | 1.2475 | Arg74, Phe358, Ile72 Met118, Met143, Val80, Glu114, Asp206, Asp150, Lys95, Ala205 | Polar hydrogen bond, Alkyl, Pi-sulfur, Pi-alkyl, Attractive charge, Halogen, Attractive charge, Pi-sigma | 2.98 (105.4°), 4.19, 3.9, 5.9, 5.2, 5.4, 5.3, 4.1, 2.9, 4.04, 3.2 | -0.427 |
| MayBridge_2787123 | -11.2772 | 1.0677 | Asp206, Phe358, Val80, Met118, Ala93, Ile72, Ala205, Leu195, Val152, Met143 | Pi-donor hydrogen bond, Alkyl, Pi-sulfur, Alkyl, Pi-sigma, Alkyl, Pi-alkyl | 3.06 (115.6°), 4.5, 3.8, 5.02, 3.5, 4.8, 3.03, 3.44, 4.08, 5.05, 5.05 | -0.451 |
| Asinex_129457932 | -11.4502 | 1.0200 | Lys190, Asp188, Leu97, Leu96, Arg74, Lys95 | Polar hydrogen bond, Pi-alkyl, Carbon hydrogen bond, Amide Pi-stacked, Pi-cation | 2.86 (103.4°), 5.11, 2.82, 5.15, 4.7 | -0.381 |
| Asinex_110075889 | -11.1804 | 1.0155 | Leu97, Asp206, Lys95, Asp107, Asp107 | Pi-alkyl, Polar hydrogen bond, Pi-cation & anion, Halogen | 5.12, 2.77 (103.4°), 5.22, 3.50, 3.23, 3.48 | -0.360 |
| Asinex_92618330 | -11.8116 | 1.0479 | Asp188, Asp206, Val80, Lys95, Lys95, Asp107, Phe77, Phe110 | Carbon hydrogen bond, Alkyl, Pi-cation & anion, Pi-alkyl | 2.3 (103.4°), 2.8 & 2.7, 5.3, 4.6, 3.9, 3.35, 5.06, 5.07 | -0.357 |

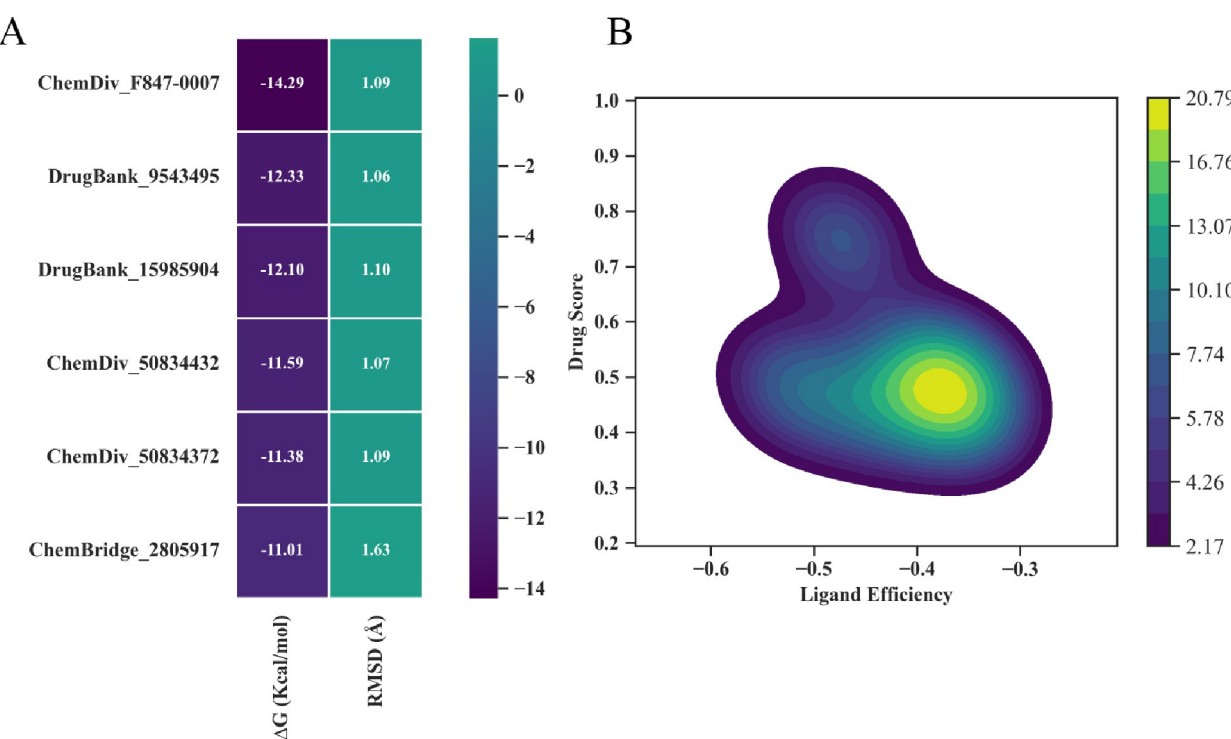

**Fig 9.** (A) Comparison of Binding affinity (ΔG) and RMSD of best six hits with the ROCK2 (B) Correlation analysis between Ligand efficiency (LE) and Drug score of best hits via Python Seaborn module.

## 3.5 DFT calculations

Quantum computational methods, frontiers molecular orbitals (HOMO and LUMO) predict stability and chemical reactivity on the basis of highest and lowest excited states (Fig 12A–12D). To understand inhibitory potential of both compounds, energy gap and other descriptors were computed (Table 4). A1 compound has 0.14735 kcal/mol energy gap with 2.005eV hardness and 0.496eV softness while A2 illustrates 0.11685 kcal/mol Eg, 1.59eV hardness and 0.628eV softness. Both compounds have shorter energy gaps and hardness along higher softness values indicating more reactivity and stability. Both compounds depict increased HOMO (donate electron) and decreased LUMO (accept electron) energies reflect the development of stable interaction and significant receptor binding. Both compounds have antioxidant ability as they show 1.87eV and 2.15eV ionization potential for A1 and A2 respectively. Electrochemical potential values for A1 (-3.875eV) and A2 (-3.74eV) revealing their charge transfer capacity to receptor while electrophilicity index classified both compounds as strong electrophiles. Detailed electrophilic and nucleophilic sites illustrated graphically by molecular electrostatic potential (MEP) which shows detailed geometries and favorable points on compounds that influence the establishment of interactions with receptor (Fig 12E and 12F). MEP highlights favorable spots via different color schemes as the negative areas in red color, positive area in blue and zero potential areas with green color.

## 3.6 MD simulation

To visualize protein-ligand stability and structural constancy of both selected (Chem-Div_F847-0007 (A1) and DrugBank_15985904 (A2)) systems and apo protein, MD simulation of 200 ns was carried out. RMSD, RMSF values of both complexes and apo protein were

**Table 2. ADMET analysis of best selected compounds along some toxicophoric rule's alert.** The prediction values transformed into six symbols: 0–0.1(—), 0.1–0.3(–), 0.3–0.5(-), 0.5–0.7(+), 0.7–0.9(++), and 0.9–1.0(+++).

| Pharmacokinetic Properties | | A1 | A2 | A3 | A4 | A5 | A6 |
|---|---|---|---|---|---|---|---|
| Absorption | Caco-2 Permeability | -4.871 | -6.158 | -7.727 | -4.913 | -4.985 | 5.254 |
| | MDCK Permeability | 1.9e-05 | 1.5e-05 | 1.4e-05 | 2.5e-05 | 2.1e-05 | 1.8e-05 |
| | skin permeation (cm/s) | -5.85 | -5.37 | -7.32 | -5.89 | -6.24 | -5.72 |
| | GI absorption | High | High | High | High | High | High |
| Distribution | BBB | No | No | No | Yes | Yes | No |
| | PPB | 100.714% | 100.249% | 76.476% | 96.361% | 97.582% | 95.849% |
| | Volume Distribution | 0.376 | 0.234 | 0.739 | 0.899 | 0.883 | 0.795 |
| | Fraction Unbound in Plasma | 0.712% | 0.692% | 19.838% | 2.628% | 2.323% | 6.902% |
| Metabolism | CYP1A2 inhibitor | ++ | + | + | - | - | +++ |
| | CYP2C19 inhibitor | +++ | + | + | +++ | +++ | +++ |
| | CYP2C9 inhibitor | +++ | ++ | ++ | ++ | ++ | +++ |
| | CYP2D6 inhibitor | ++ | + | – | ++ | +++ | ++ |
| | CYP3A4 inhibitor | +++ | – | ++ | ++ | ++ | ++ |
| Excretion | Total Clearance (ml/min/kg) | 7.514 | 0.683 | 2.091 | 3.544 | 4.869 | 1.740 |
| Toxicity | hERG Blockers | +++ | – | - | ++ | +++ | – |
| | Human Hepatotoxicity | – | +++ | ++ | ++ | ++ | +++ |
| | Drug Induced Liver Injury | ++ | +++ | +++ | ++ | ++ | +++ |
| | AMES Toxicity | + | – | – | – | – | – |
| | Rat Oral Acute Toxicity | — | - | – | ++ | ++ | ++ |
| | Eye Irritation | — | — | — | — | — | — |
| | Eye Corrosion | — | — | — | — | — | — |
| | Respiratory Toxicity | + | — | — | +++ | +++ | - |
| Toxicophoric Rules | Acute Toxicity Rule | 0 alert | 0 alert | 0 alert | 0 alert | 0 alert | 0 alert |
| | Genotoxic Carcinogenicity Rule | 0 alert | 0 alert | 5 alerts | 1 alert | 1 alert | 1 alert |
| | Nongenotoxic Carcinogenicity Rule | 0 alert | 0 alert | 1 alert | 0 alert | 1 alert | 1 alert |
| | Aquatic Toxicity Rule | 0 alert | 1 alert | 3 alerts | 0 alert | 0 alert | 1 alert |
| | Non-Biodegradable Rule | 0 alert | 1 alert | 0 alert | 0 alert | 1 alert | 1 alert |
| | SureChEMBL Rule | 0 alert | 0 alert | 0 alert | 0 alert | 0 alert | 0 alert |

enlisted in Figs 13 and 14. Ligand (A1) RMSD reached equilibrium at 12ns, and this trend continued till 200 ns. P-RMSD evaluation of A1-ROCK2 system illustrates consistent simulation throughout the time period and lie within 2.5–4.0 Å demonstrates that ligand stably bound with target protein (Fig 13A). The A2–ROCK2 system reached stability at 30ns, ligand depicted major deviations at 20ns and 120ns due to rotatable bonds and L-RMSD fluctuated around 1–5 ± 0.5 Å while P-RMSD lie within 2.2–5.0 Å with progressive upsurge at 165–200 ns (Fig 13B). Fig 13C depicts apo-protein RMSD which stabilizes at 5ns and gradually increases to 4.2 Å. These findings demonstrate that compared to apo-protein, there are no major structural differences and increase in P-RMSD value while A1 system is more stable than A2.

The observation was further supported by RMSF analysis which determines the significant rigid and flexible regions to determine its functionality and compactness. According to Fig 14A1 and 14B1, atom wise RMSD of A1 lies within 1.2–2.8 Å while L-RMSF for A2 display higher peaks within the range of 2–5 Å while both hit compounds contain 8 RBs. (Fig 14A2 and 14B2) Residue wise RMSF of A1 maintain 0.8–3 ± 0.5 Å except one higher peak and P-RMSF for A2 was less than 1–5 Å. The higher RMSF areas indicate presence of loop/coil regions while alpha helices/beta-strands make it rigid with lower fluctuations. The green

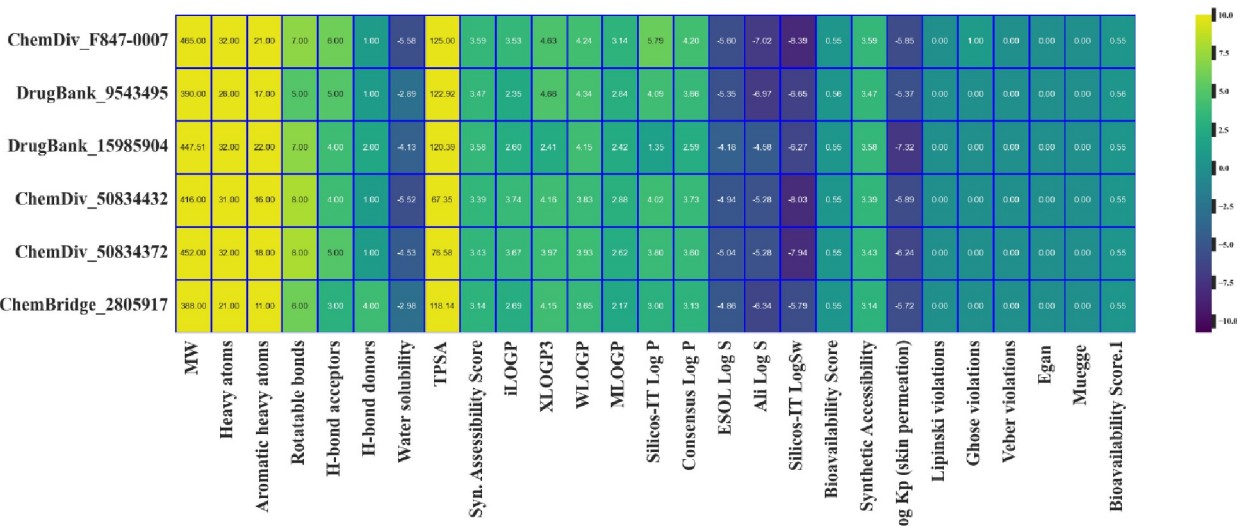

**Fig 10. Heatmap showing ADME properties of top 6 hits.** Color bar shows ADME properties ranging from low (blue) to high (yellow).

vertical bars display strong intermolecular ligand-receptor interactions with lowest fluctuations which depict high stability at the active site. Fig 14C1 illustrate RMSF for apo-protein which lies within ~4 Å while Fig 14C2 depicts 3D structure of target protein along 3 highlighted flexible regions (R1-R3). It can be observed that there is no major shift recorded in apo and selected system's RMSF, and ROCK2 maintain compactness over 200 ns simulation. Secondary structure elements monitored throughout the simulation. (S8 Fig in S1 File).

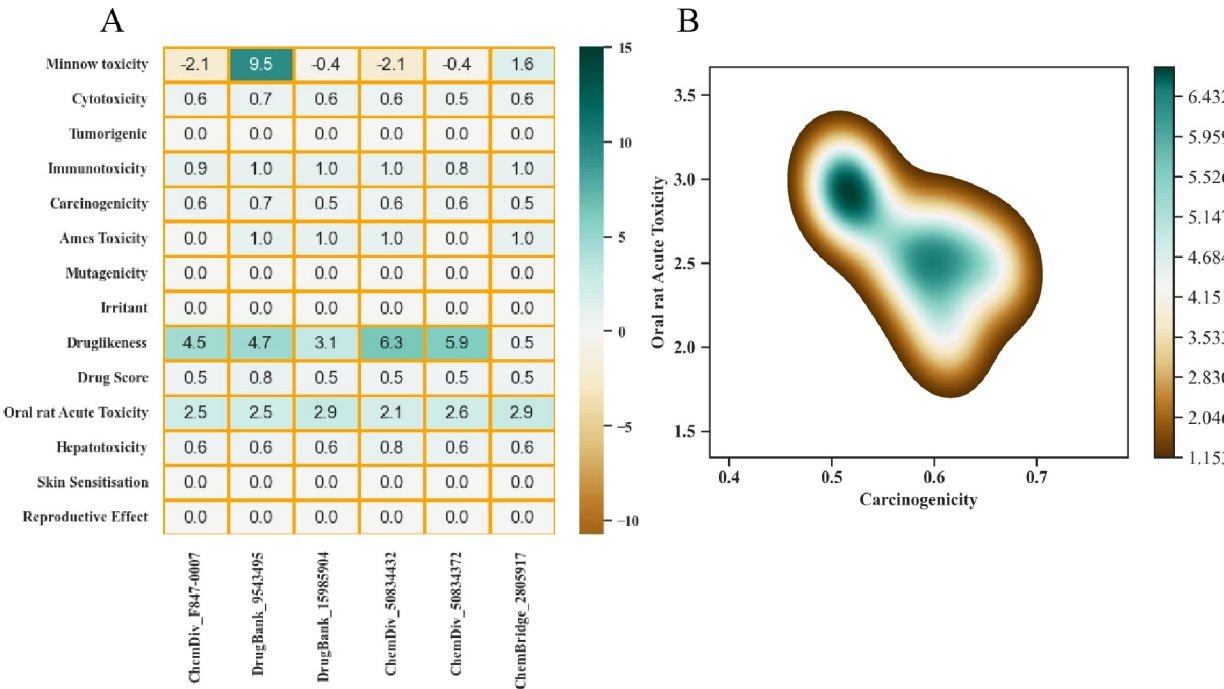

**Fig 11. Toxicity profile analysis of top six hits along KDE plot which represent joint distribution of carcinogenicity and oral rat acute toxicity by employing Python Seaborn module.** Different color intensities refer to distinct regions of data density.

Table 3. Bioactivity score of top six compounds with different human receptors.

| Compounds | GPCR ligand | Ion channel Modulator | Kinase Inhibitor | Nuclear Receptor ligand | Protease Inhibitor | Enzyme Inhibitor |
|---|---|---|---|---|---|---|
| | | | Parameters of Bioactivity Score | | | |
| A1 | -0.43 | -1.02 | -0.57 | -0.91 | -0.77 | -0.38 |
| A2 | -0.10 | -0.37 | -0.08 | -0.13 | 0.05 | -0.03 |
| A3 | 0.34 | 0.03 | 0.34 | -0.48 | -0.04 | 0.41 |
| A4 | 0.08 | 0.01 | 0.04 | -0.16 | 0.14 | -0.02 |
| A5 | 0.04 | -0.05 | 0.04 | -0.21 | 0.05 | -0.08 |
| A6 | -0.07 | -0.22 | -0.26 | -0.92 | -0.30 | 0.11 |

Secondary structure element (SSE) plot's visual analysis revealed helical nature of ROCK2 during simulation which contain 24.95% alpha helices, 13.48% beta strands, and a total of 38.43%.

Protein-ligand interaction histogram for both systems revealed many hydrophobic interactions and water bridges (Fig 15). Nonpolar aliphatic Val80, Leu195, Ala93, and aromatic

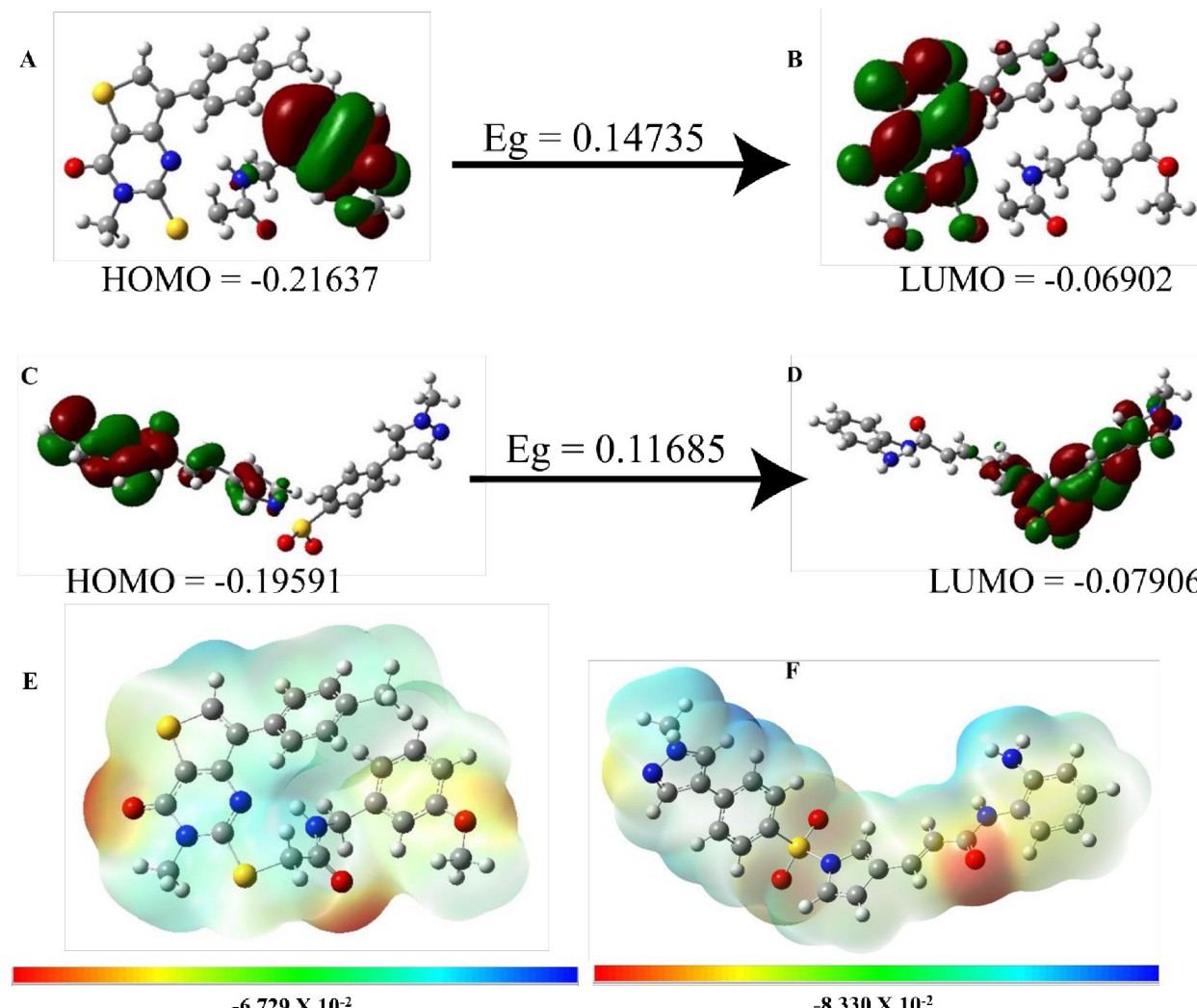

**Fig 12.** Electron density regions HOMO and LUMO (A, B) A1 compound and (C, D) A2 compound. E and F display molecular electrostatic potential of A1 and A2 compounds respectively where dark red color represents strongest electronegative region, dark blue color indicates the strongest electropositive region whereas the green color indicates zero potential.

**Table 4. DFT indices of top two compounds along calculated values of electrophilicity index, and electrochemical potential.**

| | | | | | | | | | |
|---|---|---|---|---|---|---|---|---|---|
| Parameters for DFT Analysis | | | | | | | | | |
| Compounds | HOMO | LUMO | Energy Gap (ΔE) | Ionization Potential | Electron Affinity | Electronegativity (eV) | Electrochemical Potential | Hardness (η) | Softness (S) | Electrophilicity Index (eV) |
| A1 | -0.21637 | -0.06902 | 0.14735 | 1.87 | 5.88 | 3.875 | -3.875 | 2.005 | 0.496 | 3.744 |
| A2 | -0.19591 | -0.07906 | 0.11685 | 2.15 | 5.33 | 3.74 | -3.74 | 1.59 | 0.628 | 4.398 |

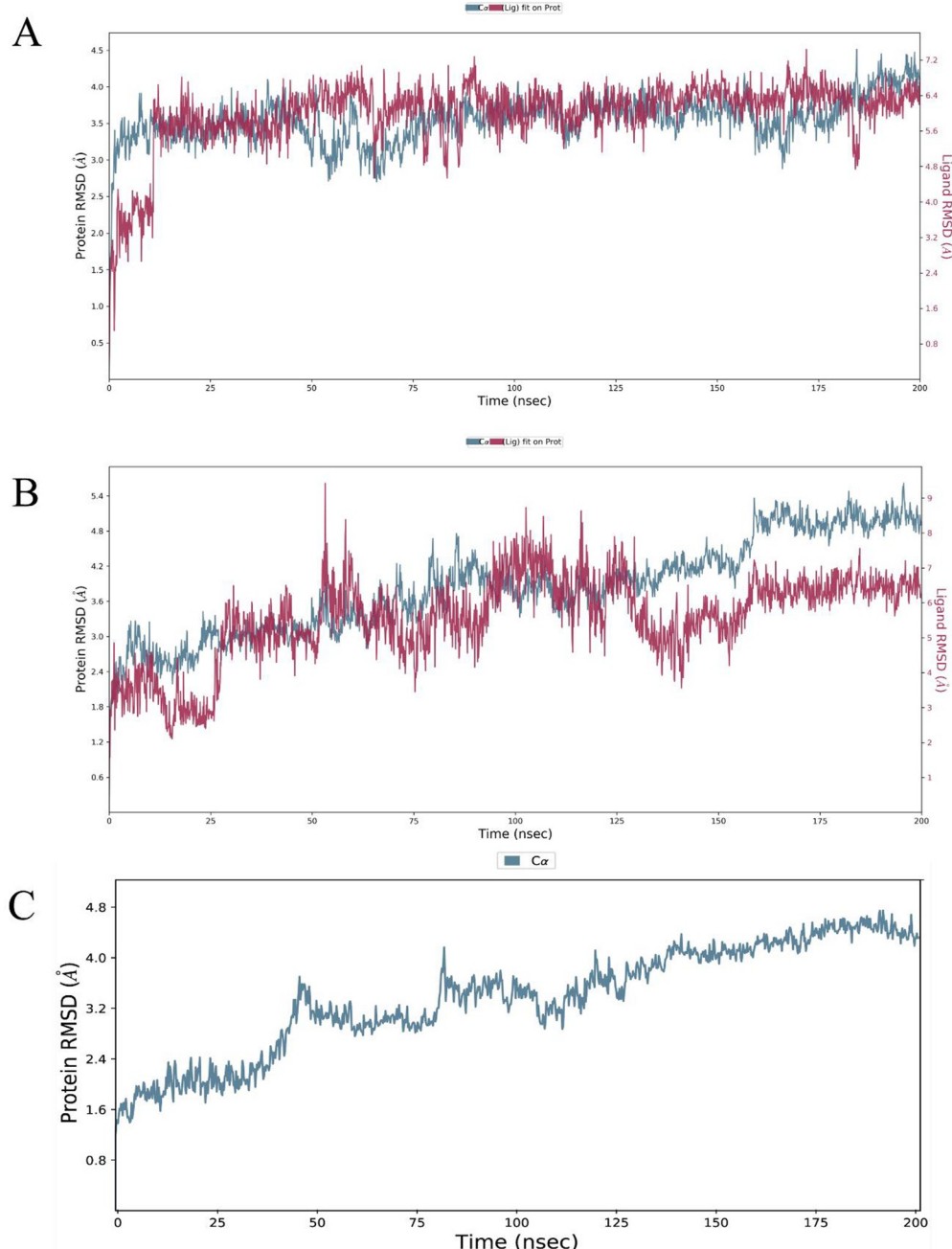

**Fig 13.** RMSD of the Cα atoms of ROCK2 and the best selected ligand A1 (A) and A2 (B) overtime. (C) RMSD of apo-protein. The left Y-axis shows variation of protein RMSD (blue line represent results) while right Y-axis show ligand variation (red lines) throughout the simulation.

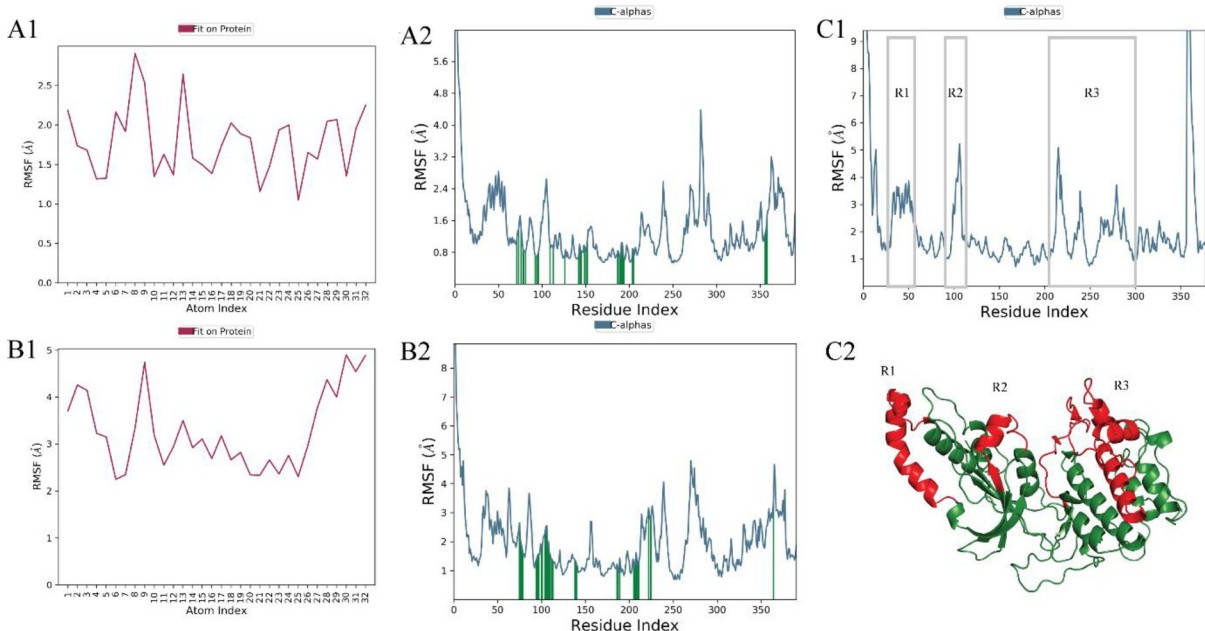

**Fig 14.** (A1) Atom-wise RMSF of A1 compound and (B1) A2 in contact with target protein ROCK2 (red color) while (A2) and (B2) shows Residue-wise RMSF of protein in contact with A1 (upper panel) and A2 (lower panel). C1 depicts RMSF of apo-protein while C2 shows the 3 most fluctuated regions during 200 ns MD simulations highlighted with red color.

Tyr145, Phe358 make hydrophobic interactions with A1 while Lys95, Ala205, Met146 make multiple interactions i.e., water bridges and H-bonds. Negatively charged Asp150 make ionic and water bridges with O13 for 60% of simulation time (Fig 15A). Aromatic Phe77, Phe110, positively charged Arg187, Lys95 display hydrophobic interactions. Negatively charged Asp107, nonpolar aliphatic Ala109 make hydrogen bonds and water bridges with N8, 9 for 32% of simulation time. Arg105, Arg187, Thr209 make multiple interactions with A2 (Fig 15B). S9 Fig in S1 File. supports the findings of histogram (Fig 15). As the protein-ligand contacts for both systems during 200 ns MD simulation displayed with orange color bands while deep orange color band depict more than one contact.

Ligand torsion profile of both systems were illuminated in S10 Fig in S1 File. The upper panel shows 2D illustration of both ligands with 8 rotatable bonds (marked with different color codes), while the lower panel depicts dials and bar plots which demonstrate ligand's conformational strain in bound state. S10A Fig in S1 File. shows 8 RBs near 180–0˚, -90˚. -70˚, 180˚, 90˚, 180˚, -90˚, -130˚ respectively. S10B Fig in S1 File. displays 4 RBs near 180˚, other 4 at 0˚, 90˚, 180˚, 90 respectively. Ligand properties were enlisted in Table 5 like RMSD was 3.0 Å and 2.0 Å for both A1 and A2 respectively. To measure the extendedness of both ligands, rGyr for both ligands were calculated which lies within 4–5.0 Å and 5.4–6.3 Å respectively. SASA was within the range of 50–200 Å$^2$ for best hit compound which stabilize at 45ns throughout the simulation but in case of second-best compound it lies within 160–320 Å$^2$. MolSA and PSA for A1 were 380–440 and 60–90 Å$^2$ and in case of A2 it lies between 412–424 and 184–200 Å$^2$ respectively.

### 3.7 Comparative conformational analysis across selected trajectories

To get insights regarding time-evolution/conformational changes as well as any significant alterations in key structural parameters, conserved and reformed ligand-amino acid bindings,

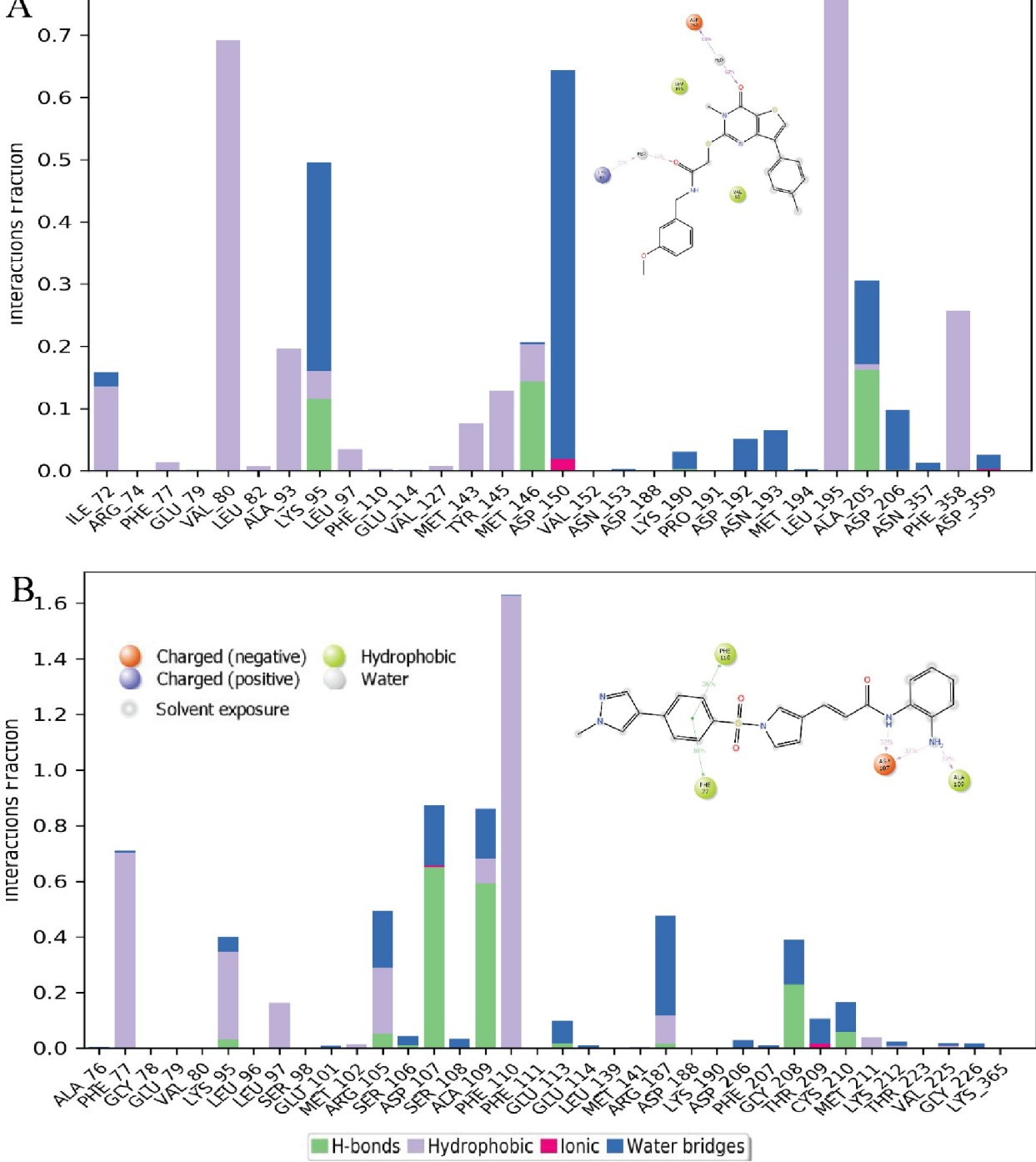

**Fig 15.** Represent interactions between protein and ligand which occur >30.0% of simulation time (A) interaction of A1 compound with ROCK2 (B) A2 and ROCK2 interactions.

compare initial middle and final frames (for 0 ns, 100 ns, 200 ns) for both ligands bound receptors (A1 and A2). Fig 16 depicts comparative conformations of A1 and A2 system at 0 ns, 100 ns and 200 ns. Notably, it's observed that between time frame 0 and 100 ns for A1 system, there is no significant alteration in 3D-structure and ligand within the ROCK2 binding site

**Table 5. Both A1 and A2 compound's properties determined via MD simulation such as intramolecular Hydrogen Bonds (intraHB), RMSD, Molecular Surface Area (MolSA), Radius of Gyration (rGyr), Polar Surface Area (PSA), Solvent Accessible Surface Area (SASA) fluctuated near equilibrated state confirm stability of both ligands.**

| Ligand properties | RMSD (Å) | RGyr (Å) | intraHB | MolSA (Å$^2$) | SASA (Å$^2$) | PSA (Å$^2$) |
|---|---|---|---|---|---|---|
| A1 (ChemDiv_F847-0007) | 3.0 | 5.2 | 01 | 440 | 200 | 90 |
| A2 (DrugBank_15985904) | 2.0 | 6.3 | 01 | 424 | 320 | 200 |

while at 200ns frame, Ala205 make significant H-bond with hinge region and cause a loss of initial H-bond with Asp206. In system A2, it is notable that residue Arg105 form H-bond occurs exclusively in 200 ns frame, while Asp107 form H-bond across all three frames. Altogether, A1 system fully occupied the active site while A2 compound demonstrates a partial displacement at 100ns frame.

### 3.8 Binding free energy landscape

Binding free energies were determined for MD trajectory frames of both systems which aid in the design and development of medication. According to Fig 17, both systems possess higher negative binding free energy values i.e., ($\Delta G_{bind}$) = -54.420 kcal/mol and -62.272 kcal/mol respectively and follow the cutoff value of -50 kcal/mol which indicate better ligand binding in receptor's active site. According to detailed energy profile, van der walls (-49.606, -49.376 kcal/mol) contribute more than other energy types while lipo (-17.64, -23.04 kcal/mol), H-bond (-1.95, -1.66 kcal/mol) and coulombic (-9.75, 2.30 kcal/mol) energy types make significant contributions. These results show consistency with earlier findings but before using them as drugs, preclinical and clinical evaluation needed towards ROCK2 receptor.

## 4. Conclusion

The current study makes use of the ROCK2 target protein and computational and medicinal chemistry methods to anticipate prospective lead compounds to inhibit ROCK2 and treat cardiovascular diseases. Pharmacophore validation followed by the high throughput virtual screening of ∼50M small molecules from different databases, top hits docked and revealed similar binding poses which were reported for reference compound. The compounds displayed excellent ADMET profile, HOMO-LUMO energies, bioavailability, and bioactivity

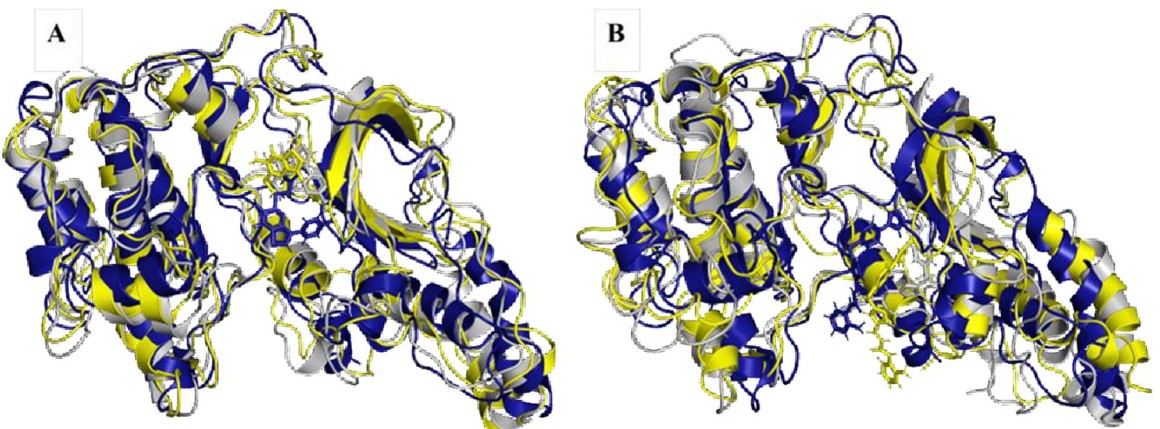

**Fig 16. Conformational analysis of (A) A1 and (B) A2 systems.** Overlays for both protein-ligand complexes at initial (0 ns) represented in blue, middle (100 ns) in yellow and final (200 ns) with grey color frames to analyze conformational changes.

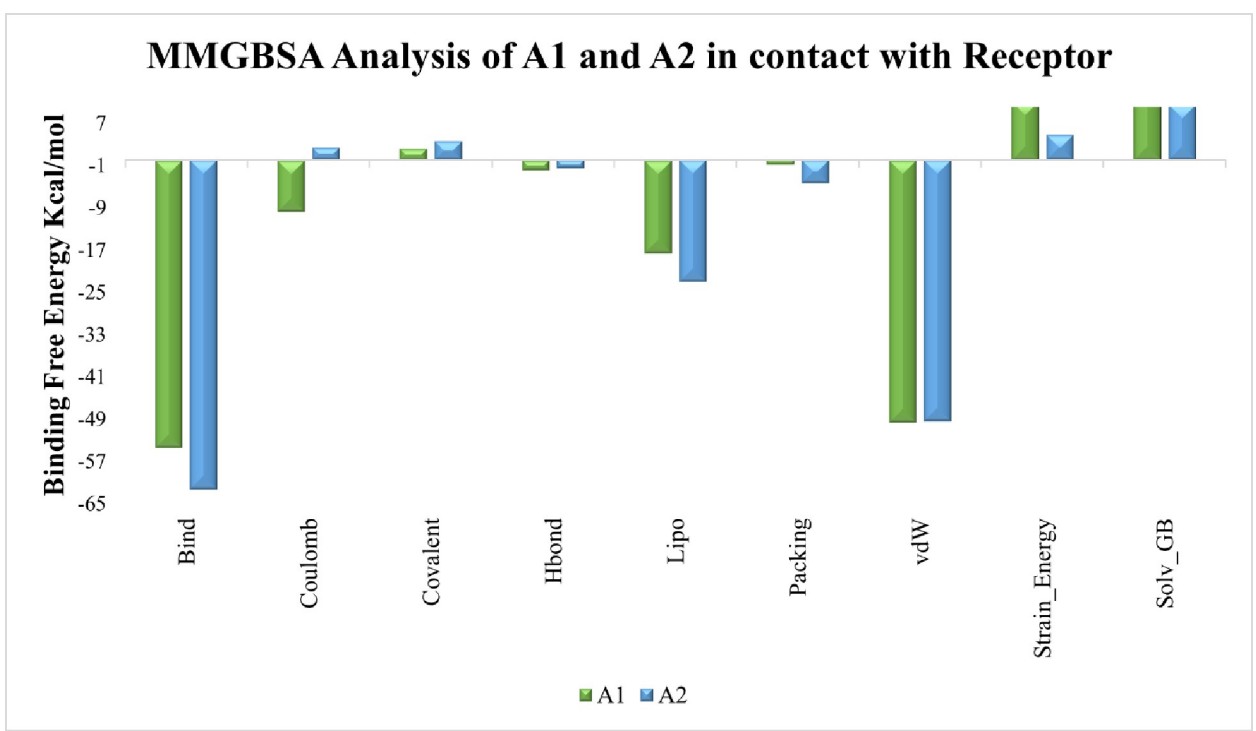

**Fig 17. Binding free energy components for A1 and A2 systems represented with green and blue color respectively.**

scores. MD simulations of top systems A1-ROCK2 and A2-ROCK2 revealed stability and didn't undergo large conformational changes. MMGBSA studies revealed strong binding affinity and retaining the activity. *In-silico* approaches help to retrieve potential compounds A1 and A2 within less time and can be used as promising drug molecules after further optimization, preclinical and clinical trials.

## Supporting information

**S1 File. Contains supporting figures and tables.**
(DOCX)

## Author Contributions

**Conceptualization:** Iqra Ali.

**Data curation:** Iqra Ali.

**Formal analysis:** Iqra Ali, Muhammad Nasir Iqbal.

**Funding acquisition:** Muhammad Ibrahim, Ihtisham Ul Haq, Wadi B. Alonazi.

**Investigation:** Iqra Ali.

**Methodology:** Iqra Ali.

**Project administration:** Iqra Ali.

**Resources:** Ihtisham Ul Haq, Abdul Rauf Siddiqi.

**Software:** Iqra Ali.

**Supervision:** Abdul Rauf Siddiqi.

**Validation:** Iqra Ali.

**Visualization:** Iqra Ali.

**Writing – original draft:** Iqra Ali.

**Writing – review & editing:** Iqra Ali, Muhammad Ibrahim, Wadi B. Alonazi.

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
