## [Decision Letter · Decision Letter 0]

16 Aug 2023

PONE-D-23-24411Computational Exploration of Novel ROCK2 Inhibitors as Precision Pharmacotherapeutics for Hypertension, Mechanistic Insights and Translational Implications in Cardiovascular Disease ManagementPLOS ONE

Dear Dr. Ali,

Thank you for submitting your manuscript to PLOS ONE. After careful consideration, we feel that it has merit but does not fully meet PLOS ONE’s publication criteria as it currently stands. Therefore, we invite you to submit a revised version of the manuscript that addresses the points raised during the review process.

We look forward to receiving your revised manuscript.

Kind regards,

Ahmed A. Al-Karmalawy, Ph.D.

Academic Editor

PLOS ONE

“Research supporting project number (RSP2023R332), King Saud University, Saudi Arabia.”

“Research supporting project number (RSP2023R332), King Saud University, Saudi Arabia. The funders had no role in study design, data collection and analysis, decision to publish, or preparation of the manuscript.”

3. Please include your tables as part of your main manuscript and remove the individual files. Please note that supplementary tables (should remain/ be uploaded) as separate "supporting information" files

Reviewers' comments:

Reviewer's Responses to Questions

**Comments to the Author**

1. Is the manuscript technically sound, and do the data support the conclusions?

Reviewer #1: Yes

Reviewer #2: Yes

Reviewer #3: Yes

2. Has the statistical analysis been performed appropriately and rigorously? 

Reviewer #1: Yes

Reviewer #2: Yes

Reviewer #3: I Don't Know

3. Have the authors made all data underlying the findings in their manuscript fully available?

Reviewer #1: No

Reviewer #2: Yes

Reviewer #3: Yes

4. Is the manuscript presented in an intelligible fashion and written in standard English?

Reviewer #1: Yes

Reviewer #2: Yes

Reviewer #3: Yes

5. Review Comments to the Author

Reviewer #1: Authors of the presented manuscript introduced computational approach for systematically down-selected chemical library compounds for repurposing against ROCK2 biotarget. Combined ligand and structure-based screening through pharmacophoric modelling, Molecular docking analysis, ADMET/drug-likeness profiling, DFT analysis, and molecular dynamics simulations have highlighted promising molecular reactivity and binding affinity of two identified hits. The manuscript is comprehensive, well-written, and considered valuable within its field. Few suggestions and comments are presented:

1. In line 107, ROCK2 abbreviation should be introduced earlier within the manuscript as well as within the abstract section.

2. In Figure 3, It is advised that the authors provide differential 3D-conformation for the active and inactive ROCK2 states for further illustrations as well as allow readers to track the descriptive information regarding the ROCK2 domains/regions across lines 107-118.

3. Aligned training compounds for generating the pharmacophoric model should presented in Figure 6 or at least within a supplementary figure.

4. Section 3.2. Authors provided comparative data regarding ligand binding modes through both highlighted polar hydrogen bonds and hydrophobic contacts. However, hydrogen binding should be presented within hydrogen bond distances as well as bond angles since hydrogen bond depend on both. Authors should mention the Hydrogen bond angles as well as their distances, since the strength of hydrogen bonding is based on both parameters in a way to ensure the adequacy of optimum hydrogen bonding.

5. In Figures 9 and 11, the software adopted for generating the bivariate frequency plots should be annotated at the figure legends.

6. I did not have the opportunity to check the manuscript tables, despite the authors’ annotation that these tables are within a “Table file”, which is actually missing from the submission link.

7. For the RMSD and RMSF calculations, authors are advised to illustrate trajectories for apo protein as well. This approach would better highlight the impact of compound’s binding on target through pinpointing flexible and immobile patterns for the protein ternary structures and amino acids in reference to the unliganded form.

8. Regarding the ROCK2’s RMSF figures, these figures should be highlighted with the zones for the main secondary structures, loops, and motifs being involved within the target activation and/or reported substrate binding. This would provide comparative insights regarding the ligand binding impact on the important target substructures and domains.

9. Authors are advised to provide overlays for the initial, middle, and final frames (at 0 ns, 100 ns, and 200 ns, respectively) for each ligand-protein complex across the molecular dynamics simulations. This approach would provide great insights regarding the time-evolution orientation/conformation changes for both the protein and bounded ligands as well as the conserved and reformed ligand-amino acid bindings and close-range contacts.

10. Authors should elaborate more on the discussion section through presenting comparative findings from reported literature studies that investigated reported compounds against the same target protein.

11. Additionally, within the discussion sections, authors should highlight the takeaway messages that would be adopted in future lead optimization and development based on the molecular docking, DFT analysis, ADNET profiling, and MD simulations. Prospective/recommended structure modifications to improve the ligand’s binding and interactions, as well as pharmacokinetics should be provided within the discussion and conclusion sections.

12. Minor typos should be corrected like “H-bons” at line 443.

Reviewer #2: Dear Authors

This manuscript entitled as "Computational Exploration of Novel ROCK2 Inhibitors as Precision Pharmacotherapeutics for Hypertension, Mechanistic Insights and Translational Implications in Cardiovascular Disease Management" by Iqra Ali1, et al reported a detailed Insilico analysis of a wide array of molecules and identified their potential in ROCK2 inhibition. The study is framed in a nice manner and the obtained results are quite interesting to the respective field. However, I would like to suggest some refinement to the current form of manuscript, before acceptance for publication as follows.

1. Title of the manuscript is needing some sort of reframing. Without proper in-depth biological evaluation or translational studies, the present title is unsuitable or misleading. Kindly consider to revise the title according to the experiments reported in the present manuscript. For instance, according to “Precision Pharmacotherapy: Integrating Pharmacogenomics into Clinical Pharmacy Practice” by Hicks etal, “Precision pharmacotherapy encompasses the use of therapeutic drug monitoring; evaluation of liver and renal function, genomics, and environmental and lifestyle exposures; and analysis of other unique patient or disease characteristics to guide drug selection and dosing”. In this perspective, authors should justify each terminology in the title. A similar scenario can be seen in the case of “translational implications”. Kindly consider a revision in the title.

2. The language used in the manuscript needs to be improve in the manuscript. I would like to suggest a thorough revision in the overall language used. For instance, the spelling mistakes in the abstract. … Therefore, using Insilco techniques predict novel…… “I” is missing the insilico.

3. ….. 2000 best hits were selected and apply rule of 5, then docked in the binding site of ROCK2 to understand interactions of ROCK2 with hit compounds… ROCK2 is repeating in the same line.

4. In the abstract section, the only the methodology is reported. Authors should include the major results in the abstract.

5. Use of abbreviations and their expansion. Whenever using an abbreviation, kindly include the expansion on the first time use of abbreviations. For instance, …. ROCK2 target due to its substantial role in NO-CGMP and RhoA/ROCK pathway…. Authors should use this sort of methods throughout the manuscript. Though list of abbreviations are included, some places abbreviations along with the expansion is used in the manuscript. Kindly make a uniform pattern.

6. In the introduction section, first line is not clear to the reader. Kindly try to simplify the bigger statement to the simpler one.

7. … According to WHO report, there is 29% mortality rate due to CVDs… The present data is reported to the Pakistan demography or throughout the world. This is not clear from the manuscript.

8. … dysfunction of NO-cGMP pathway….. Needs to add the expansion. Kindly check this for all abbreviations.

9. It is better to add the expansions under the legend of Figure 1.

10. …. For the cure of cardiovascular diseases….. Authors should use an alternative term for cure. “Management” of the cardiovascular diseases is more suitable than the term “cure”. Kindly consider a revision.

11. Authors described some ROCK inhibitors such as fasudil, Y27632 and ripasudil in the manuscript. Are these molecules are included in the virtual screening? What is its effects in comparison to the most potential analogues in the manuscript?

12. Does any specific ROCK1 pdb structures are available? If so, what is the difference between ROCK1 v/s ROCK2.

13. …….Dual ROCK1 and ROCK2 inhibitors have been linked to problematic side effects as well as some investigators…. Kindly discuss the side effects too. Further, is it a side effects or an adverse effect? Kindly differentiate the difference between side effect v/s adverse effect.

14. Advantage/benefit over ROCK1 needs to include for a proper understanding.

15. Authors needs to discuss about the cocrystallised ligands and their interactions with the target along with their potential role. This will enable an easier comparison with the potential analogues.

16. Grid validation for the docking protocol is needs to be included in the manuscript. RMSD variation should also be included.

17. Kindly include the units of each parameters in the manuscript. For instance, in some places, the Kcal/mol is missed along with the binding score. Kindly check this for all cases.

18. Authors cross cited some tables in the manuscript. However, there are no tables in the manuscript. Kindly check this.

19. Can authors make a preliminary structural feature analysis observation made by the most active analogue and compare the same with the currently reported molecules.

Reviewer #3: COMMENTS TO AUTHOR:

This manuscript describes the computational Exploration of Novel ROCK2 Inhibitors as Precision Pharmacotherapeutics for Hypertension, Mechanistic Insights, and Translational. In this study, the author performed Molecular docking, DFT, pharmacokinetic, and molecular dynamics studies and identified lead compounds against ROCK2 to cure cardiovascular diseases. The article is interesting and organized, but some minor concerns are there.

1. The introduction section is too lengthy for specific cardiovascular diseases, We suggest omitting some of the sentences from the introduction.

2. On page no 1, In silico word should be italic font. Check-in all places in the manuscript.

3. On page 2, line 64 …Author mentioned that According to WHO report, there is a 29% mortality rate due to CVDs….But which year of the WHO report …..?

4. On page 6, lines 147 and 156…What is the acronym of PI and UCSF….?

5. On page 7, line 168 … IC50 should be IC50

6. On page 8, line 241…Check space

7. In Figure 5 A and B are there, but in Figure 5 figure legends are missing which one is A and B

8. In Figure 7 Top hit compounds are in clinical trials..? or approved what is the medicine category and include somewhere in the text

6. PLOS authors have the option to publish the peer review history of their article (what does this mean?). If published, this will include your full peer review and any attached files.

Reviewer #1: No

Reviewer #2: No

Reviewer #3: No

---

## [Author Response · Author response to Decision Letter 0]

16 Oct 2023

Reviewer 1

1: We have carefully considered your comment regarding the abbreviation of the "ROCK2" in the abstract section. We appreciate your suggestion. We have made the necessary revisions to the abstract to introduce the "ROCK2" abbreviation at line 37 and provide readers with a clear understanding of the key target of our study. This change not only enhances the abstract's clarity but also contributes to a seamless reading experience for our audience.

2: We greatly appreciate your insightful comment. After a comprehensive search of the Protein Data Bank (PDB) and related resources. The desired information related to the ROCK2 domains/regions has been incorporated in section 01. The Figure 3 details general functional mechanism of the protein instead of Angstrom level molecular details. The key residues implicated in the active and inactive ROCK2 protein have been discussed in detail with high resolution description in results and discussion section. 

3: Thank you for your valuable feedback and your keen interest in our work. We appreciate your suggestion regarding the presentation of aligned training compounds used to generate the pharmacophoric model.

We incorporate your suggestion by including a figure of the top aligned training compounds as Figure S4. This addition will help readers visualize and understand the basis on which the pharmacophore model was constructed.

4: Thank you for your insightful comment and detailed attention to our work. We truly appreciate your valuable feedback regarding the bond distance and angles. Bond distance is already presented in 3D depiction and in Table 1 while H-bond angles are added in Table 1. Via both hydrogen bond distances and bond angles, we are able to provide a more complete picture of the hydrogen bonding interactions within our analysis which will contribute to a better understanding of the nature and strength of the hydrogen bonds formed between the ligands and the protein. 

5: Respected sir, thank you for your valuable feedback and attention. To generate heatmap and plots Python Seaborn module was employed. We added this information in figure legends. Your comment underscores the value of thoroughness in our manuscript, and we address this aspect to enhance the scientific rigor of our study.

6: Thank you for bringing this to our attention. We apologize for any inconvenience you've encountered in accessing the manuscript tables. We promptly investigated the issue and ensured that the "Table file" was provided in the submission link. This time, we added all tables at the end of the revised manuscript file. so you will be able to thoroughly review the manuscript tables.

7: We would like to express our gratitude for your constructive comment to illustrate trajectories for the apo protein alongside our RMSD and RMSF calculations is indeed valuable for providing a comprehensive view of the impact of the compound's binding on the target.

We incorporate this recommendation into our revised manuscript. Specifically, trajectory plots for the apo protein in addition to the ternary structures, RMSD, and RMSF plots (line 434-437 and 449-453). This addition will allow readers to visually compare the dynamic behavior of the protein in its unliganded form with that in the presence of the compound. By showcasing the trajectories of the apo protein, we aim to better elucidate the flexible and immobile patterns of the protein, highlighting how they change upon ligand binding. This will provide a more detailed understanding of the structural dynamics induced by the compound, contributing to the overall clarity and depth of our analysis.

8: Regarding your comment concerning the RMSF (Root Mean Square Fluctuation) figure, we wholeheartedly agree that providing a more detailed and informative representation of the protein's structural dynamics can enhance the clarity and impact of our findings. 

In response to your feedback, we make the following enhancements to the RMSF Figure 14 in our manuscript:

Structural Annotations: We annotate RMSF figures to indicate the locations of key secondary structures, specially loops and substrate binding regions.

Color Coding: To improve clarity, we highlight flexible and mobile regions with red color while rest with green colored cartoon, making it easier for readers to identify specific regions of interest.

Additional Explanatory Text: We include explanatory text in the figure captions to provide context and briefly describe the functional significance of the annotated structural elements.

These improvements will allow readers to gain a more comprehensive understanding of how ligand binding impacts specific substructures of ROCK2, providing valuable insights into the molecular mechanisms at play.

9: We incorporate overlays for the specified time points in the molecular dynamic simulations. These overlays illustrate the time-evolution orientation and conformational changes for both the protein and the bound ligands. Furthermore, we emphasize the conserved and reformed ligand-amino acid bindings and close-range contacts, providing readers with a clearer understanding of the dynamic behavior and interactions.

10: I appreciate your valuable comment. To address this point, we expand the interaction analysis section to incorporate comparative findings of co-crystal complexes (investigated compounds targeting the same protein). This comparative analysis allows us to draw connections between our findings and those of other researchers, and potential implications.

11: Thank you for your valuable comment. Future lead optimization and development efforts for hit compounds typically involve a series of strategic steps such as structural optimization, bioavailability enhancement, In vitro and In vivo toxicity assessments and efficacy in relevant disease model as preclinical evaluation. Further is to ensure patent protection for novel lead compounds and conduct clinical trials. These steps enhance the compounds' drug-like properties and maximize their potential as therapeutics. Changes made in the manuscript. 

12: Thank you respected sir for your thorough review. We appreciate your keen eye in identifying minor typographical errors. We promptly correct this typographical error and any other minor typos that may have inadvertently crept into the manuscript during the drafting and editing process. Our commitment to delivering a polished and error-free manuscript remains paramount.

Reviewer #2

1: Your guidance in this matter is appreciated. We revise the title to better encapsulate the key elements of our research, ensuring that it is a more accurate representation of the study's content.

2: We acknowledge the need for enhanced language precision and do a thorough revision of the manuscript to address the issues. Rectify spelling errors, ensuring grammatical correctness, and improving the clarity of expressions throughout the manuscript.

3: Thank you for your careful review. For clarity purposes, we revise the sentence in the manuscript and eliminate the repetition of ROCK2.

4: Thanks for your comment. Your guidance is instrumental in improving the clarity and comprehensibility of our manuscript. We make the necessary revisions to the abstract to ensure that it includes a brief but informative overview of the major results obtained in our study. We recognize that this will enhance the abstract's completeness and better convey the significance and final results of our research to readers. 

5: Thank you for your constructive feedback. To make a uniform pattern, exclude list of abbreviations and add expansion on the first-time use of abbreviations throughout the manuscript.

6: Thank you sir for your valuable suggestion. We rephrase in the revised manuscript to make it more concise and reader-friendly while retaining the essential context and message.

7: We appreciate your observation. To clarify this, the presented data is for worldwide. We will make this explicit in the manuscript text to ensure there is no confusion about the geographic scope of the data.

8: In revised manuscript, we added all abbreviations, including "NO-cGMP pathway," especially expanded upon their first usage in the text to provide readers with a clear understanding of their meaning and maintaining consistency in regard to enhance the manuscript's comprehensibility.

9: We certainly take your recommendation into consideration and include the expansion directly under the legend of Figure 1. 

10: Thank you for your attention to detail. We promptly revise the manuscript to replace term "cure" with "treatment" to accurately convey the intended meaning.

11: We appreciate your keen interest and thanks for your comment. Fasudil, Y27632, and ripasudil were not included in the virtual screening that led to the identification of the most potential analogues presented in our manuscript. Our primary objective in the virtual screening was to identify novel compounds with potentially enhanced activity compared to known inhibitors. These ROCK inhibitors employed during pharmacophore modelling (extracted important features by aligning all template ligands). In section 3.2, we include the most significant interactions that co-crystalized ligands made with target protein and their effects to the potential hits.

12: Yes, specific ROCK1 structures (PDB ID: 6E9W) are available in protein data bank. ROCK1 and ROCK2 contain kinase domains with 90% identity, coiled coil domain (55% identity), and PH domain (65% identity) as their overall structures are quite similar with distinct functions and roles in cellular processes. They are involved in various signaling pathways and cellular functions. The specific functions and downstream effectors of ROCK1 and ROCK2 vary. They also differ in their tissue distribution. Upon alignment of ROCK1 (PDB ID: 6E9W) and ROCK2 (PDB ID: 6ED6), they depict 85% overall alignment score.

13: Dual ROCK1 and ROCK2 inhibitors can lead to a drop in blood pressure, causing symptoms such as dizziness, Fatigue, Headache, Myalgia and fainting. Inhibition of ROCK1 and ROCK2 can affect cardiac function, potentially leading to bradycardia (slow heart rate) or arrhythmias (irregular heart rhythms). Nausea, vomiting, diarrhea, and abdominal pain are common gastrointestinal side effects associated with dual ROCK1 and ROCK2 inhibitors. Some individuals experience peripheral edema as side effect of dual inhibitor. In most cases, they cause side effects while in certain conditions lead towards adverse effects.

Side effects are the secondary, usually undesired effects of a drug. such as nausea, dizziness, or drowsiness. Adverse effects refer to undesirable and unintended harmful effects resulting from the normal use of a medication or medical intervention. These effects can be mild to severe.

14: Though both isoforms of ROCK regulate phosphorylation of myosin light chain and MLCP activity, but ROCK1 is not directly bound to MLCP’s myosin-binding subunit named MYPT1 while ROCK2 bind directly. When ROCK2 bind with MYPT1 it starts phosphorylation of MLC and cause contraction instead of relaxation. In this perspective ROCK2 is known as best target for cardiovascular diseases. As well as both isoforms are not functionally redundant when controlling various aspects of the activity of myosin. 

Both ROCK1 and ROCK2 have different subcellular localization due to which they expressed differently across various tissues. Except brain and muscle, ROCK1 is widely expressed in liver, lung, kidney, spleen, testis. While ROCK2 is abundantly expressed mostly in brain, heart, muscle, and placenta. ROCK2 plays a vital function in the contractility of smooth muscle cells than ROCK1. ROCK2 gene single nucleotide polymorphisms (SNP) are related to coronary artery diseases and the hypertension. All these results confirm the crucial role of ROCK2 in cardiovascular diseases. Already discussed after figure 1 in manuscript.

15: Thank you for your valuable comment. In response to your suggestion, we include the most significant interactions that co-crystalized ligands made with target protein in section 3.2. This comparative analysis will enable to highlight similarities and differences, providing a deeper insight into the potential of analogues in our research.

16: Thanks for your valuable comment. Grid validation for a docking protocol involves assessing the accuracy and reliability of the grid that defines the binding site within a molecular docking study. Validation done via ROC curve which depicts false positive and true positive fractions on x-axis and y-axis respectively (Figure S6). Based on receiver operating characteristic curve (ROC curve), 0.853 AUC (area under curve) value was observed, and enrichment factor in top 1% was observed 8.88. The results suggest that MOE did not produce false-positive results. RMSD variation added in Table 1. Changes made in section 3.2.

17: Thank you sir, for this valuable feedback on our manuscript. We thoroughly review the manuscript to ensure that units are consistently provided for all relevant parameters, including binding scores. We made changes in the manuscript.

18: Thank you for your comment and for bringing this issue to our attention. We apologize for any inconvenience you've encountered in accessing the manuscript tables. We promptly investigated the issue and ensured that the "Table file" was provided in the submission link. This time, we added all tables at the end of the revised manuscript file. So, you will be able to thoroughly review the manuscript tables.

19: Thank you for your insightful comment. To address your valuable comment, we mention detailed structural feature analysis of the most active compound and compare it with hit molecules (section 3.2). Also, describe significant interactions of active ligands with ROCk2 and make comparison with 2 best hits (Figure S5). At the time of pharmacophore modelling, we also analyzed the structural features: one oxygen as hydrogen bond acceptor, one nitrogen as hydrogen bond donor and two aromatic rings. 

Reviewer #3

1: We acknowledge the importance of conciseness in scientific writing, but we believe that the comprehensive introduction provided in our manuscript is essential to establish the necessary context and background for our study. 

Upon re-evaluation, we have determined that each sentence and section in the introduction contributes significantly to the overall understanding of the research, and no content can be omitted without compromising the clarity and completeness of the context. Therefore, we have decided to retain the current structure and content of the introduction section.

2: Thank you for this correction. In response to your suggestion, we ensure that the term "In silico" is consistently formatted in italic font throughout the manuscript. 

3: Thanks for drawing our attention towards WHO report year specification. The mortality rate due to CVDs can indeed vary over time due to advances in healthcare and changes in population demographics.

To address this comment, we immediately review and verify the most up-to-date WHO report on CVD mortality rates and include the specific report year in our manuscript. We understand the importance of providing accurate and current data to support our findings and conclusions. We appreciate your diligence in ensuring the accuracy and credibility of our work, and your feedback significantly contributes to the quality of our manuscript.

4: The acronym "pI" stands for "isoelectric point," of protein. It is a critical parameter in biochemistry and molecular biology, particularly in the context of protein characterization and electrophoresis while "UCSF," stands for the University of California, San Francisco, which is a prominent research institution who developed chimera. UCSF Chimera or Chimera used for interactive visualization and for modelling purposes.

5: Thanks for this correction. Updated in the manuscript.

6: Thank you for your attention to detail and for bringing the formatting issue to our attention. We appreciate your careful review of our manuscript.

Upon your suggestion, we have reviewed page 8, specifically line 241, and have identified and corrected the spacing issue. We understand the importance of maintaining consistent formatting and clarity in our manuscript.

7: We appreciate your diligence in reviewing our manuscript. Your feedback is instrumental in improving the quality and clarity of our work. You are correct; there is an oversight in Figure 5 where the figure legends for panels A and B are missing. We apologize for this oversight, and in the revised manuscript we promptly address this issue.

8: These are predicted hits from computational analysis and on the basis of Insilico results these hits have promising properties for inhibiting ROCK2 while to check their biological activity and specific medicine categories we have to push the best hits towards In vitro study and further subjected towards clinical trials.

---

## [Decision Letter · Decision Letter 1]

25 Oct 2023

PONE-D-23-24411R1Computational exploration of novel ROCK2 inhibitors for cardiovascular disease management; insights from high-throughput virtual screening, molecular docking, DFT and MD simulationPLOS ONE

Dear Dr. Ali,

Thank you for submitting your manuscript to PLOS ONE. After careful consideration, we feel that it has merit but does not fully meet PLOS ONE’s publication criteria as it currently stands. Therefore, we invite you to submit a revised version of the manuscript that addresses the points raised during the review process.

We look forward to receiving your revised manuscript.

Kind regards,

Ahmed A. Al-Karmalawy, Ph.D.

Academic Editor

PLOS ONE

Journal Requirements:

Reviewers' comments:

Reviewer's Responses to Questions

**Comments to the Author**

1. If the authors have adequately addressed your comments raised in a previous round of review and you feel that this manuscript is now acceptable for publication, you may indicate that here to bypass the “Comments to the Author” section, enter your conflict of interest statement in the “Confidential to Editor” section, and submit your "Accept" recommendation.

Reviewer #1: All comments have been addressed

Reviewer #2: (No Response)

Reviewer #3: All comments have been addressed

2. Is the manuscript technically sound, and do the data support the conclusions?

Reviewer #1: Yes

Reviewer #2: Yes

Reviewer #3: Yes

3. Has the statistical analysis been performed appropriately and rigorously? 

Reviewer #1: Yes

Reviewer #2: N/A

Reviewer #3: Yes

4. Have the authors made all data underlying the findings in their manuscript fully available?

Reviewer #1: Yes

Reviewer #2: Yes

Reviewer #3: Yes

5. Is the manuscript presented in an intelligible fashion and written in standard English?

Reviewer #1: Yes

Reviewer #2: Yes

Reviewer #3: Yes

6. Review Comments to the Author

Reviewer #1: (No Response)

Reviewer #2: Authors have improved the manuscript in a nice manner. However, I would like suggest some minor comments.

1.Please pay close attention to both spelling and the use of expansions. In some places, it is written as 'In-Silico,' while in others, it appears as 'insilico.' Let's ensure uniformity by carefully reviewing the entire manuscript.

2.A similar situation can be observed with 'NO-CGMP.' Should the 'C' be in lowercase or uppercase? Please review and make consistent adjustments throughout the document.

3.Please exercise caution when using the word 'cure.' Once a person is affected by cardiovascular diseases, it typically remains with them for life. I am curious to know if there is a possibility of curing the condition with medication. To the best of my knowledge, patients need to continue taking medication throughout their lifespan for the management of condition. In this context, the use of 'cure' is inappropriate. Please correct me if I am mistaken.

4.I am delighted to see that standard drugs such as fasudil, Y27632, and ripasudil have been employed in the preparation of pharmacophore modeling. However, I am curious about the effects of these molecules in the docking experiments. Including these results would facilitate a better understanding of the potential by comparing various parameters. I suggest incorporating this study into the manuscript.

5.The representation of 'kcal/mol' is incorrect in many parts. Please review and update accordingly.

6.The units for RMSD values are missing in the table.

7.Similarly, the units for distance are also missing.

Reviewer #3: The author has duly considered and responded to all the comments. The manuscript is suitable for publication.

7. PLOS authors have the option to publish the peer review history of their article (what does this mean?). If published, this will include your full peer review and any attached files.

Reviewer #1: **Yes**

Reviewer #2: No

Reviewer #3: No

---

## [Author Response · Author response to Decision Letter 1]

31 Oct 2023

Response Letter 

To

Emily Chenette and Ahmed A. Al-Karmalawy,

Editor in Chief and Academic Editor,

PLOS ONE 

Respected Mis/Sir,

Thank you for your kind coordination, and thanks to the reviewers for their constructive suggestions in enhancing the quality and clarity of our research. We have addressed the reviewers and corrected as per the suggestions. Please find the list below of point-by-point responses to the reviewer. Please let me know if you have any further suggestions. 

Corresponding authors

Dr. Abdul Rauf Siddiqi

Dr. Iqra Ali

Point-by-Point Response:

Reviewer: 2

Authors have improved the manuscript in a nice manner. However, I would like to suggest some minor comments.

1. Please pay close attention to both spelling and the use of expansions. In some places, it is written as 'In-Silico,' while in others, it appears as 'insilico.' Let's ensure uniformity by carefully reviewing the entire manuscript.

Response: Thank you for your valuable suggestions. We appreciate your attention to detail. We carefully review the entire manuscript to ensure uniformity in both spelling and the use of expansions. We make necessary corrections for a consistent and polished final document. Your input is instrumental and improves the quality of our work.

2. A similar situation can be observed with 'NO-CGMP.' Should the 'C' be in lowercase or uppercase? Please review and make consistent adjustments throughout the document.

Response: Certainly, the comment's concern regarding the capitalization of 'C' in 'NO-cGMP' has been noted at some points. To maintain consistency throughout the document, the 'C' changed to lowercase, and all instances of 'NO-CGMP' adjusted to 'NO-cGMP.' Thank you for your observation.

3. Please exercise caution when using the word 'cure.' Once a person is affected by cardiovascular diseases, it typically remains with them for life. I am curious to know if there is a possibility of curing the condition with medication. To the best of my knowledge, patients need to continue taking medication throughout their lifespan for the management of condition. In this context, the use of 'cure' is inappropriate. Please correct me if I am mistaken.

Response: Certainly, it's essential to use precise terminology when discussing medical conditions and treatments. You're correct that the term 'cure' may not be the most appropriate when addressing cardiovascular diseases. However, we use the word 'treat' instead of cure. 'Treatment' more accurately reflects the ongoing care required to manage cardiovascular diseases, and I appreciate your input in ensuring accurate communication in this context. Thank you for the suggestion.

4. I am delighted to see that standard drugs such as fasudil, Y27632, and ripasudil have been employed in the preparation of pharmacophore modeling. However, I am curious about the effects of these molecules in the docking experiments. Including these results would facilitate a better understanding of the potential by comparing various parameters. I suggest incorporating this study into the manuscript.

Response: We'd like to inform you that we already compared the results of our docking experiments with standard compound 3SG (S6 Fig.) but now upon your request we dock all three enlisted standard drugs i.e., Fasudil, Y27632. However, considering the amount of data and the potential complexity, we made the decision to include the results (S5 Fig.) in the supplementary material while a bit description in the manuscript. This approach was taken to maintain the clarity and flow of the primary manuscript, while still providing interested readers with access to the additional data for a more in-depth analysis. We believe that this arrangement allows for a comprehensive examination of our findings without overcomplicating the main text.

5. The representation of 'kcal/mol' is incorrect in many parts. Please review and update accordingly.

Response: Thank you for pointing out the error in the representation of 'kcal/mol'. We reviewed the content and made the necessary corrections to ensure accuracy.

6. The units for RMSD values are missing in the table.

Response: The missing units for RMSD values in the table have been added. Thank you for bringing this to our attention.

7. Similarly, the units for distance are also missing.

Response: To address this concern, the distance units have been included. Thank you for bringing this to our notice, and we appreciate your comments.

---

## [Decision Letter · Decision Letter 2]

3 Nov 2023

Computational exploration of novel ROCK2 inhibitors for cardiovascular disease management; insights from high-throughput virtual screening, molecular docking, DFT and MD simulation

PONE-D-23-24411R2

Dear Dr. Ali,

We’re pleased to inform you that your manuscript has been judged scientifically suitable for publication and will be formally accepted for publication once it meets all outstanding technical requirements.

Kind regards,

Ahmed A. Al-Karmalawy, Ph.D.

Academic Editor

PLOS ONE

Reviewers' comments:

Reviewer's Responses to Questions

**Comments to the Author**

1. If the authors have adequately addressed your comments raised in a previous round of review and you feel that this manuscript is now acceptable for publication, you may indicate that here to bypass the “Comments to the Author” section, enter your conflict of interest statement in the “Confidential to Editor” section, and submit your "Accept" recommendation.

Reviewer #2: All comments have been addressed

2. Is the manuscript technically sound, and do the data support the conclusions?

Reviewer #2: Yes

3. Has the statistical analysis been performed appropriately and rigorously? 

Reviewer #2: N/A

4. Have the authors made all data underlying the findings in their manuscript fully available?

Reviewer #2: Yes

5. Is the manuscript presented in an intelligible fashion and written in standard English?

Reviewer #2: Yes

6. Review Comments to the Author

Reviewer #2: Authors have addressed the concern in a nice manner, and it may be accepted for publication in its current form.

7. PLOS authors have the option to publish the peer review history of their article (what does this mean?). If published, this will include your full peer review and any attached files.

Reviewer #2: No

---

## [Editor Report · Acceptance letter]

9 Nov 2023

PONE-D-23-24411R2 

Computational exploration of novel ROCK2 inhibitors for cardiovascular disease management; insights from high-throughput virtual screening, molecular docking, DFT and MD simulation 

Dear Dr. Ali:

I'm pleased to inform you that your manuscript has been deemed suitable for publication in PLOS ONE. Congratulations! Your manuscript is now with our production department. 

Kind regards, 

on behalf of

Dr. Ahmed A. Al-Karmalawy 

Academic Editor

PLOS ONE